# A small area model to assess temporal trends and sub-national disparities in healthcare quality

Adrien Allorant [1,2,3] ✉, Nancy Fullman[2,3], Hannah H. Leslie[4], Moussa Sarr[5], Daouda Gueye[5], Eliudi Eliakimu [6], Jon Wakefield [7], Joseph L. Dieleman[3,8], David Pigott [3,8], Nancy Puttkammer[9] & Robert C. Reiner Jr [3,8]

Monitoring subnational healthcare quality is important for identifying and addressing geographic inequities. Yet, health facility surveys are rarely powered to support the generation of estimates at more local levels. With this study, we propose an analytical approach for estimating both temporal and subnational patterns of healthcare quality indicators from health facility survey data. This method uses random effects to account for differences between survey instruments; space-time processes to leverage correlations in space and time; and covariates to incorporate auxiliary information. We applied this method for three countries in which at least four health facility surveys had been conducted since 1999 – Kenya, Senegal, and Tanzania – and estimated measures of sick-child care quality per WHO Service Availability and Readiness Assessment (SARA) guidelines at programmatic subnational level, between 1999 and 2020. Model performance metrics indicated good out-of-sample predictive validity, illustrating the potential utility of geospatial statistical models for health facility data. This method offers a way to jointly estimate indicators of healthcare quality over space and time, which could then provide insights to decision-makers and health service program managers.

Subnational differences in healthcare capacity and delivery contribute to inequities in health outcomes[1], and thus monitoring such patterns is critical for promoting better health for all populations. However, few - if any - data systems optimally measure access and provision of high-quality healthcare, over time or across locations[2]. In their absence, health facility surveys, including the Service Provision Assessment (SPA) and the Service Delivery Indicators (SDI) surveys, remain primary data sources for assessing indicators of health service availability and quality in low- to middle- income countries (LMICs)[3–8]. These surveys,

conducted in over twenty countries across sub-Saharan Africa, Latin America and the Caribbean, Southern Asia, and Eastern Europe, provide detailed information about service components and capacities, grouped into structures (basic amenities, infection control, equipment, diagnostics, and medication), processes (components of clinical care), and outcomes, including patients' satisfaction with services received[9].

To summarise quality of care from health facility surveys, two metrics are generally used[5,10–13]: readiness, which measures the

[1]Department of Epidemiology, Biostatistics, and Occupational Health, McGill University, Montreal, QC, Canada. [2]Department of Global Health, University of Washington, Seattle, WA, USA. [3]Institute for Health Metrics and Evaluation, University of Washington, Seattle, WA, USA. [4]Division of Prevention Science, University of California San Francisco, San Francisco, CA, USA. [5]Institut de Recherche en Santé de Surveillance Epidémiologique et de Formation, Dakar, Senegal. [6]Health Quality Assurance Unit, Ministry of Health, Dodoma, Tanzania. [7]Department of Statistics and Department of Biostatistics, University of Washington, Seattle, WA, USA. [8]Department of Health Metrics Sciences, School of Medicine, University of Washington, Seattle, WA, USA. [9]International Training and Education Center for Health (I-TECH), Department of Global Health, University of Washington, Seattle, WA, USA. ✉e-mail: adrien.allorant@mail.mcgill.ca

availability of functioning physical resources (e.g., equipment, essential medicines, diagnostic capacities) and staff in assessed facilities[14]; and process quality, which measures providers' compliance with accepted standards of care.

Recent work has highlighted large subnational disparities in the coverage of key health services[15], interventions[16] and outcomes[17,18], in LMICs, underscoring the need to better understand and monitor subnational variations in the readiness and process quality of health services provided[19]. Furthermore, the devolution of health service provision from the national to the district-level has substantially increased local governments' responsibilities in the planning and implementation of public health[20–23], including the maintenance and equipment of health facilities, which has generated an enhanced demand for subnational indicators[24].

This study presents a Bayesian mixed modelling approach using publicly available facility surveys and covariates data, to assess changes in the readiness and process quality of facility-based health services at a subnational scale. Our approach has two steps (Fig. 1): first, metrics of readiness and process quality are aggregated at the subnational level using facilities' survey weights, a common method for bias removal when analysing complex surveys[25–27]; second, subnational area-level readiness and process quality metrics are modelled as a function of space-time smoothing processes, spatially and temporally indexed covariates, and survey-type random effects.

Previous analyses of health facility data have reported cross-sectional survey estimates of readiness and process quality by administrative area[7,28] or survey-year[29–31], but differences in sample size and design pose challenges to comparability across locations and time,

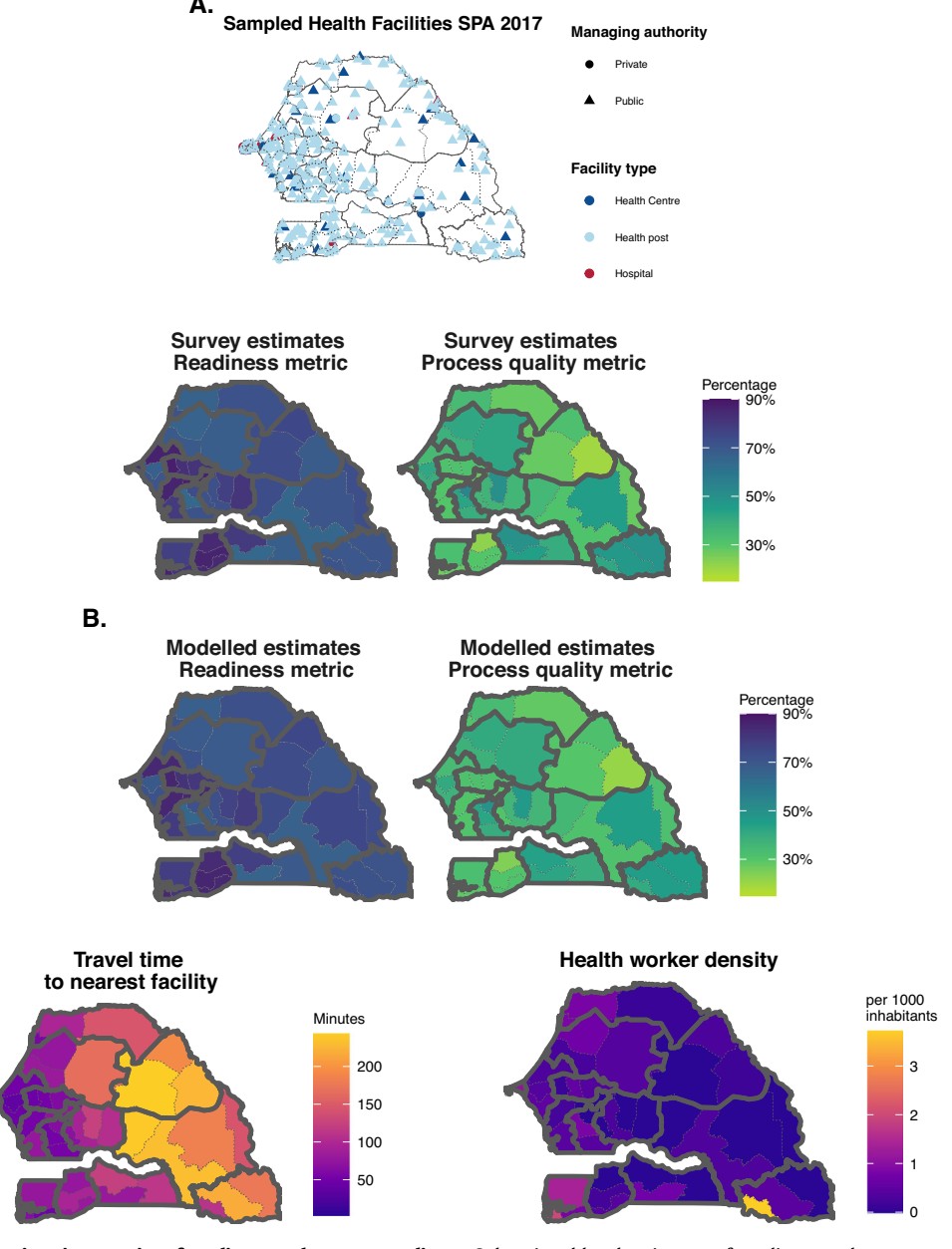

**Fig. 1 | Modelling steps for estimating metrics of readiness and process quality subnationally and over time. A** Step 1: in each country, metrics of readiness and process quality are calculated in sampled facilities (**Sampled health facilities** map), and then aggregated using facilities' survey weights to the subnational level (**Survey estimates Readiness and Process quality** maps). **B** Step 2:

Subnational-level estimates of readiness and process quality (top maps) are obtained from a model using space-time smoothing and spatially referenced covariates (exemplified with bottom maps). Data for Senegal in 2017 are shown for reference here. SPA = Service Provision Assessment survey. Source data are provided as a Source Data file.

and no estimates are available between survey-years. Our approach explicitly accounts for the differences between survey instruments and design, leverages correlation over space and time, and supplements direct survey measurements with auxiliary information from space-time-indexed covariates, to produce time-series of quality metrics at programmatic resolution. To evaluate the model, we perform hold-one-area-out cross-validation; mean error, mean absolute error, and coverage are calculated to assess bias, precision, and calibration of the predictions. We apply this approach to three countries in which at least four health facility surveys have been conducted since 1999 – Kenya, Senegal, and Tanzania – and produce yearly estimates of metrics of sick-child care quality per WHO guidelines by county, department, and region, respectively.

## Results
### Model performances
In areas and years where survey data were available, we compared survey and modelled estimates; the best small area models produced, in each country, area-level estimates of readiness and process quality metrics close to the direct survey estimates (Fig. 2). The difference between modelled and survey estimates were greatest among areas where the survey estimates had lower precision (i.e., greater variance). This was expected as in areas where direct survey estimates were less reliable, the model was designed to draw more information from neighbouring areas, years, and auxiliary covariates, while in areas where the survey provided estimates with high precision, the model mostly reproduced the survey estimate.

In Kenya (Fig. 2A, B), agreement between survey and modelled county-level estimates was generally high and highest in 2018; this corresponds with the SDI 2018 survey, which had the largest number of observations (i.e., over 3,000 facilities and > 4 times more any other survey in Kenya). In Senegal (Fig. 2C, D), discrepancies between survey and modelled department-level estimates were largest in 2018, especially with modelled readiness metric estimates substantially larger than their survey counterparts. This is likely since survey estimates were available in 2017 and 2019, the model tends to reconcile spatial patterns to obtain smooth trends over time, which here meant mitigating the drop observed in the survey in 2018 in several areas. In-sample comparisons showed the largest differences between survey and modelled estimates in Tanzania. In particular, the discrepancies seem largest for the regional estimates of the readiness metrics derived from the SDI 2014 survey, which partially overlapped with the SPA 2014-15, a survey with a much larger sample size (~4 times larger). Thus, in reconciling the spatial patterns observed in both surveys, the model downweighed the SDI 2014 regional estimates compared to the SPA 2014-15 estimates.

To assess the validity of models' predictions for areas and years where no data were collected, we used out-of-sample cross-validation (see Methods section for details). We summarised the small area models' performances across healthcare quality metrics using bias, precision, and calibration measures- the mean error and mean absolute error (Fig. 3A), and coverage (Fig. 3B). For the readiness metric model, the mean absolute error and mean error were 3.7% and −0.7% in Kenya, 4.4% and 0.1% in Senegal, and 3.6% and −1.7% in Tanzania. Mean absolute error and mean error tended to be slightly larger for the process quality metric model, with 5.3% and −2.5% in Kenya, 5.3% and −0.7% in Senegal, and 5.0%, and 0.0% in Tanzania. Nevertheless, these values indicated a high degree of out of sample predictive accuracy.

Coverages were highest for both metrics in Kenya and Tanzania, with levels higher than their nominal values of 50%, 80 and 95%. This may be due to wider uncertainty intervals in these two countries where we modelled healthcare quality metrics over long time periods; wider modelled uncertainty increases the likelihood that the survey estimate would fall within the uncertainty intervals. Conversely, in Senegal, model coverages were, for both metrics, consistently lower than their

nominal level of 50%, 80 and 95%, suggesting that while central predictions presented low bias and high precision, the predictive distributions were not perfectly calibrated to the observed survey estimates.

### Comparison of survey and modelled estimates of metrics
Comparing subnational modelled and survey-based estimates for readiness and process quality shows how the model can generate spatial patterns found in the survey data (Fig. 4, with the example of departments in Senegal for 2019). Survey and model estimates of metrics' mean, and uncertainty intervals overlap closely. For Senegal in 2019, survey mean estimates fall within the 95% model prediction interval in all but one department for the readiness metric, and two departments for the process quality metrics. However, in these three cases, the survey-based 95% confidence intervals span 0-100%, indicating that few to no observations were available in these departments for that year, suggesting that these surveys' mean estimates are not reliable. Another striking feature of this comparison is that, in most departments, especially those with few observations, the model 95% posterior prediction intervals are significantly narrower than the survey 95% confidence intervals. This highlights the added precision gains that can be obtained from a model leveraging information across space and time. We present similar results for Kenya, and Tanzania in Supplementary Results 3.5.

### Examples of model outputs
We illustrate the model usage with three examples of model outputs: (i) maps of areal mean estimates of metrics with associated uncertainty; (ii) within-area variability in estimates of metrics by facilities' managing authority, and (iii) by facilities' level of care, in Senegal (equivalent model outputs for Kenya and Tanzania are included in Supplementary Results 3.6).

Estimates can be used to visualise spatial patterns in healthcare quality metrics (Fig. 5). For instance, mean estimates of the readiness metric were estimated to be highest in the Western departments of Senegal (Top panel), while the process quality metric was estimated to display more subnational heterogeneity (Bottom panel). The uncertainty plots (Fig. 5- right panel), which show mean estimates against the width of their associated 95% uncertainty interval, are important reminders that apparent subnational heterogeneity in estimates can reflect meaningful differences or substantial uncertainty in the estimates. For instance, there is substantial uncertainty in the estimates of the process quality metric, for most of the Northern and Eastern departments in Senegal, where few health facilities were sampled and the total number of sick-child consultations in the data was small.

Area-level estimates of readiness and process quality metrics can vary between facilities depending on their managing authority or their level of care. Figure 6A illustrates uniformly higher estimates of the mean readiness metric in public facilities compared to private facilities in Senegal, across departments. The estimated average process quality metric varied substantially across facility types- for instance, among hospitals, we estimated high levels of the process quality metric in the southern departments but low levels in the south-eastern departments (Fig. 7B).

Stratified analyses were however associated with substantially higher mean absolute error and mean error, and lower coverage than the non-stratified analyses, suggesting lower precision, higher bias, and poorer calibration (Fig. S5).

## Discussion
Monitoring subnational healthcare quality is critical for identifying and addressing geographic inequities in service provision. With this study, we developed a spatial analytic approach to generate estimates of facility-based readiness and process quality of care at a subnational resolution. Our Bayesian hierarchical model supports the

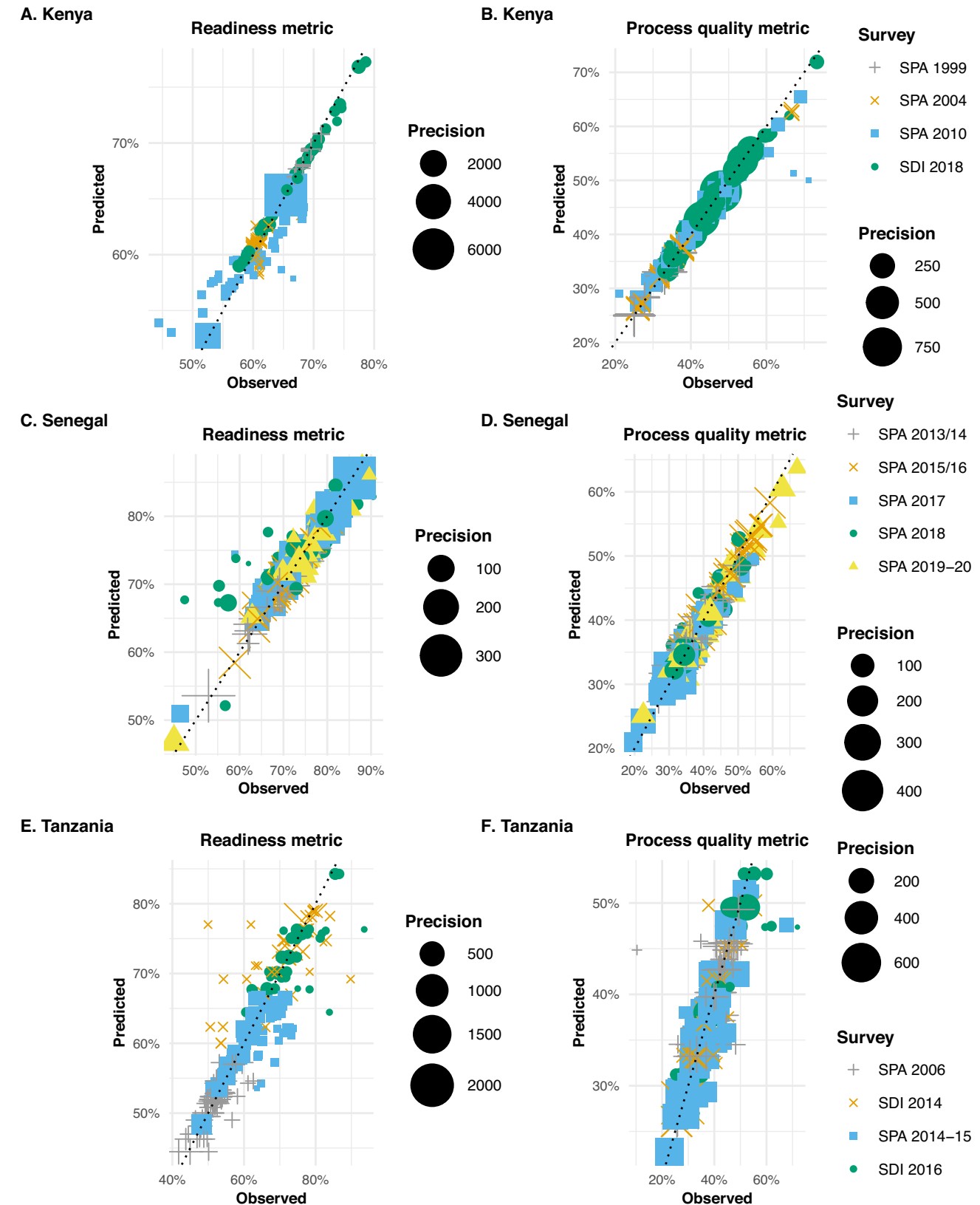

incorporation of multiple sources of health facility survey data, while accounting for the specific design and uncertainty of each data collection instrument. When recent health facility surveys are not available, our approach leverages space-time smoothing and auxiliary information from geo-referenced covariates to produce subnational annual estimates of readiness and process quality. This method offers

a replicable approach for generating subnational and temporal estimates for healthcare quality indicators. Furthermore, our approach can be used to critically assess the alignment of health facility assessment tools and the metrics derived from them.

Our study expands on previous work on healthcare quality measurement in several ways. Because differences in surveys' design and

**Fig. 2 | In-sample validation results for estimated readiness and process quality metrics, at the area-level.** In-sample validation results for estimated readiness (left panel) and process quality (right panel) metrics, at the area-level, in Kenya (**A** and **B**), Senegal (**C** and **D**), and Tanzania (**E** and **F**). The dotted line has a slope of 1, showing the relationship between survey-estimated and model-estimated readiness and process quality metrics. Points' colours and shapes represent the different survey-years from which the direct survey estimates were derived. For instance, in Kenya (Fig. 2A, B), green dots represent county-level estimates from the SDI survey conducted in 2018, while blue rectangles represent county-level esti- mates derived from the SPA survey conducted in 2010. As facility surveys were not

all powered to produce reliable estimates of healthcare quality metrics at fine spatial resolution, we want to distinguish on the plot between area-level survey estimates with higher precision (i.e., lower variance) and less reliable area-level survey estimates (with lower precision). For each metric and country, points are sized based on area-level survey estimates' precision (i.e., the inverse of the var- iance of the survey estimates). For instance, in Kenya, two county-level estimates of the readiness metric derived from the SPA 2010 show high precision- large blue rectangles, while most others show low precision (Fig. 2A). SPA = Service Provision Assessment survey; SDI = Service Delivery Indicators survey. Source data are pro- vided as a Source Data file.

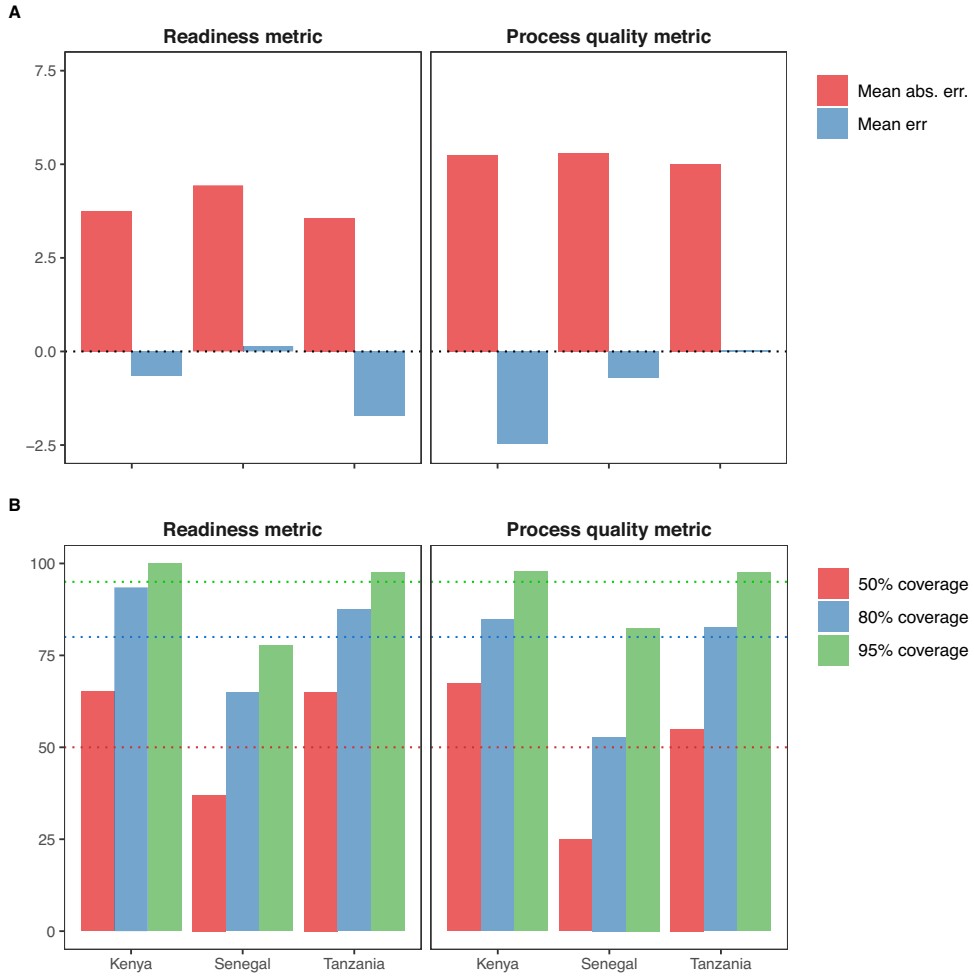

**Fig. 3 | Measures of bias, precision, and calibration of models' predictions.** Measures of bias and precision (Panel **A**), and calibration (Panel **B**) of models' predictions, using hold-out predictions of readiness and process quality metrics, in subnational areas of Kenya, Senegal, and Tanzania. Mean error, mean absolute error, and coverage were calculated across all administrative areas, using cross validation. Hold-out predictions are obtained by removing out all observations in an area and year when fitting the model and predicting the average readiness or

process quality metric with uncertainty. We repeat this process for all area-year with survey estimates; because facility surveys are not powered to produce reliable estimates at fine spatial resolution, we limit this analysis to area-year where the variance of the survey estimate is below the 50th percentiles of all areal estimates' variance, by metric, country, and year. Dotted lines on panel **B** represent the 50%, 80 and 95% coverage nominal levels. Mean err = Mean error; Mean abs. err. = Mean absolute error. Source data are provided as a Source Data file.

sampling frames limit the comparability of healthcare quality metrics estimates, studies to date tended to analyse facility surveys individu- ally. We present a two-stage approach that directly addresses this challenge by incorporating survey sampling weights and explicitly modelling potential systematic differences between surveys. In the first stage, sampling weights, which are derived from the sampling frame, are incorporated to calculate direct survey estimates of metrics, to account for the fact that health facilities have vastly different probabilities of inclusion in the survey. For instance, public hospitals are overrepresented and private clinics underrepresented in most SPA and SDI surveys. If healthcare quality metrics are associated with

managing authority or levels of care, ignoring sampling weights would lead to bias direct estimates. In the second stage, survey-specific adjustments are added to account for observed discrepancies between survey estimates due to systematic differences across measurement instruments. By allowing to jointly analyse multiple surveys from the same or different data collection tools, the proposed approach enables to look at temporal trends in subnational estimates healthcare quality metrics. Drawing from the small area estimation literature, we could envision a unit-level model as an alternative to the two-stage area-level approach presented here[32]. A unit-level approach models outcomes at the level of the primary sampling unit- for example, clusters of

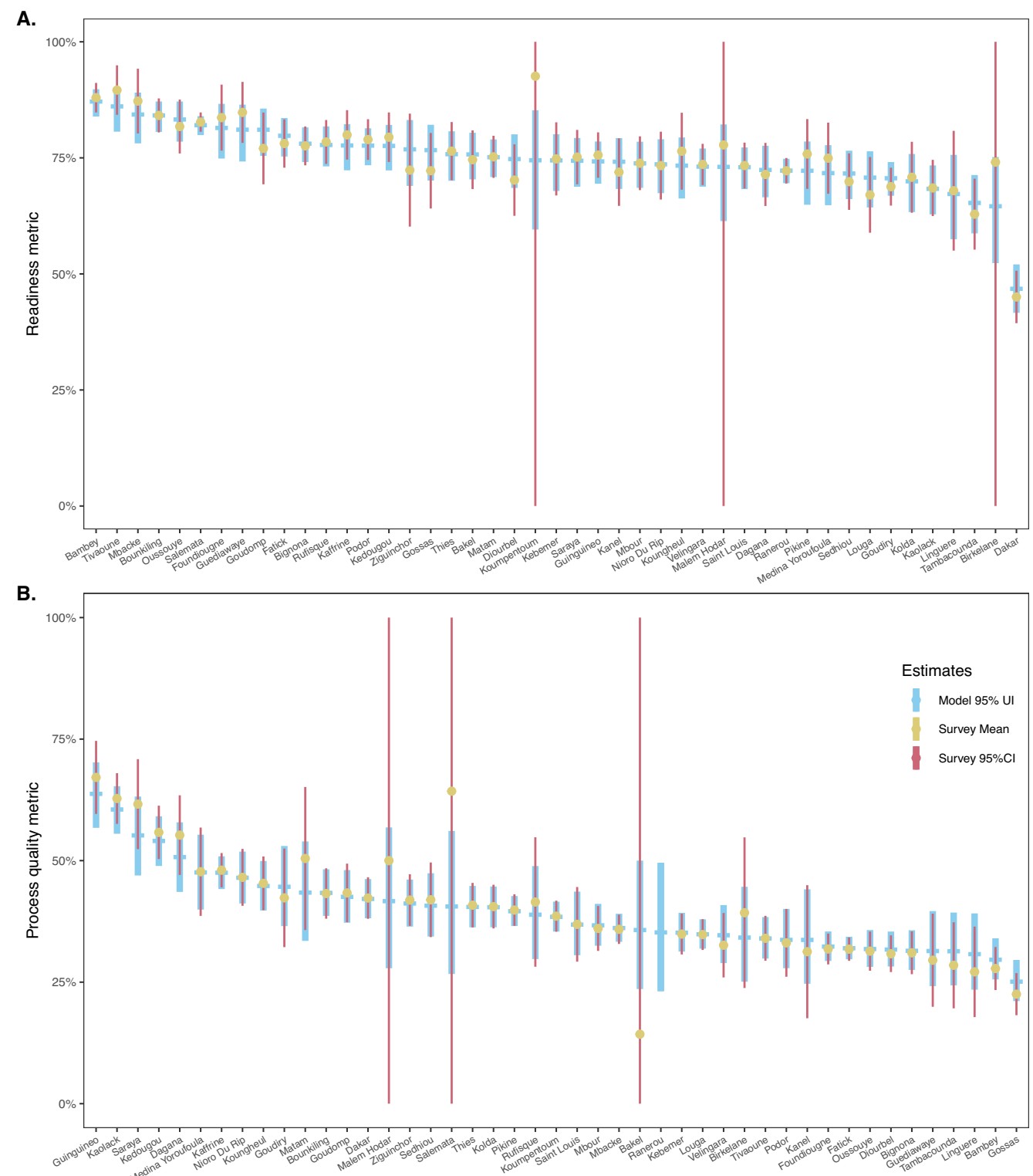

**Fig. 4 | Comparison of department-level survey and model estimates for the readiness and process quality metrics.** Comparison of department-level survey and model estimates for (Panel **A**) the readiness and (Panel **B**) process quality metrics in Senegal, in 2019. This figure compares empirical survey and model estimates, for the most recent survey-year in Senegal. Thick light-blue dash and vertical ranges show model posterior mean estimates, and the 95% posterior prediction intervals. Yellow dots and narrow red vertical lines indicate survey estimates and 95% confidence intervals, derived from SPA 2019 ($n = 361$ facilities sampled, panel **A**; $n = 885$ consultations observed in 253 facilities, panel **B**). Source data are provided as a Source Data file.

households in demographic and health surveys. With this approach, healthcare quality metrics would be modelled at the facility-level directly, and the complex survey design acknowledged by including survey strata as covariates in the model. As stronger spatial correlations might exist at smaller spatial scales, this unit-level approach would have the potential to unveil clustering or disparities at a finer resolution than the area-level model- for instance, between neighborhoods, or urban centers and rural peripheries. Estimating

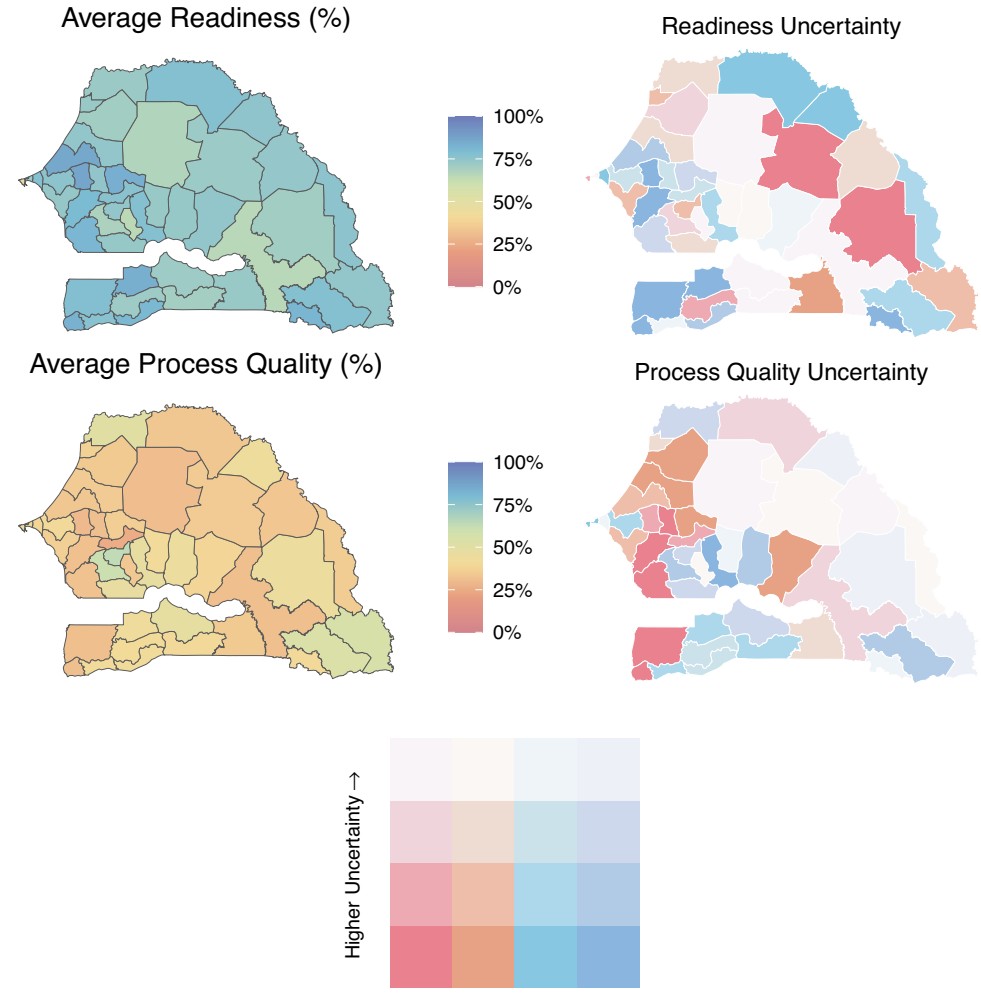

**Fig. 5 | Maps of model-estimated readiness (top panel) and process quality (bottom panel) metrics by subnational areas in Senegal in 2020, with associated uncertainty.** The left panel presents estimates of the mean, while the right panel shows both estimates of the mean and their associated 95% uncertainty interval width. Mean estimated metrics are split into quartiles; the cut-off points indicate the metric estimates' minimum, 25th, 50th, and 75th percentiles, and maximum, which were 46.8%, 72.2%, 74.5%, 77.8%, and 87.1%, for the readiness metric, and 25.1%, 33.7%, 38.4%, 43.4%, and 63.7%, for process quality. The confidence intervals' width minimum, 25th, 50th, and 75th percentiles, and maximum, were 4%, 8.6%, 10.5%, 12.2%, and 26%. Source data are provided as a Source Data file.

healthcare quality metrics across subnational areas using a unit-level approach would however be challenging. Aggregation to area-level estimates of metrics would indeed require weights to combine estimates of healthcare quality metrics from different strata; in the case of SPA surveys, these weights would correspond to the proportion of public/private hospitals, health centres, and clinics out of all formal facilities, in each subnational area. Although databases of geolocated health facilities are increasingly made available[33], existing georeferenced facility lists cannot be used as censuses of all formal health facilities. WHO's Geolocated Health Facility Data initiative may however greatly improve the completeness of health facility master lists, which would render the unit-level modelling approach truly feasible[34].

Secondly, our study underscores the benefits of using a model-based approach, which can incorporate multiple data sources, to supplement survey measurements with covariate data. Space-time smoothing can help inform model estimates in data-sparse contexts[35], and auxiliary covariate information can help parse out the space and time patterns related to metrics of interest[36]. Additionally, in years or regions where no survey data is collected, a model-based approach can build from space-time correlation and covariate data to provide estimates of metrics with associated uncertainty. Thus, direct estimates

derived from surveys that are typically conducted irregularly can be supplemented with readily available and continuously collected, spatially and temporally indexed covariates. Nevertheless, most of these covariates are environmental data derived from satellite imagery, and few seem relevant to healthcare quality[37]. However, the small area estimation approach presented here only requires that some of all the auxiliary covariates contain information about healthcare quality metrics to improve the precision of estimates[38]. Hold-one-area-out cross validation can be used to appraise the bias, precision, and calibration of the model's predictions. While we found low mean error and mean absolute error for the subnational analyses using all facilities, stratified analyses focusing on one managing authority or level of care could lead to substantial biases and poor calibration. Data availability seems to pose a major constraint on stratifying observations by subnational areas and by facility type, as extremely low counts in several areas can lead to unreliable estimates. Thus, if differences in healthcare quality metrics between public/private facilities or hospitals/clinics and health centres are an important goal of the analysis, we recommend assessing sequentially the feasibility of producing stratified estimates- from the more aggregated to the finer spatial resolution.

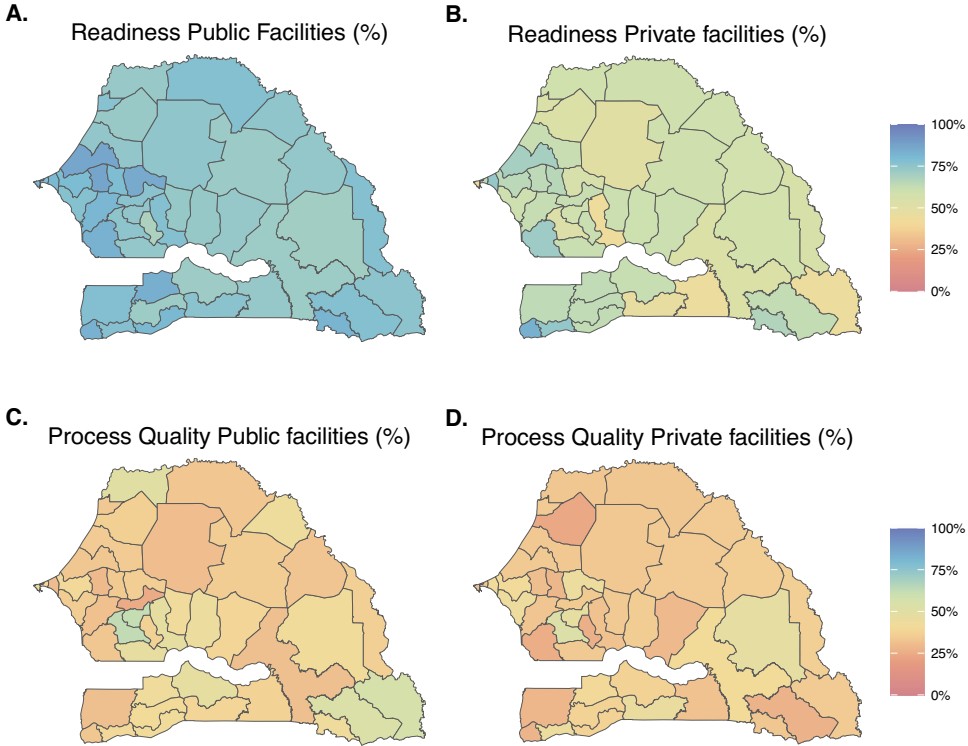

**Fig. 6 | Maps of model-estimated readiness (top panel) and process quality (bottom panel) metrics by subnational areas, and managing authorities, in Senegal in 2020.** Panels **A** and **B** (respectively **C** and **D**) are maps of modelled area-level estimates of readiness (respectively process quality) for analyses stratified on public and private facilities. Source data are provided as a Source Data file.

Thirdly, our modelling approach can be used to critically assess the production of healthcare quality metrics. To begin with, it can help inform the optimal frequency and scope of health facility assessments, which to date have been conducted as occasional surveys (Kenya and Tanzania), one-time census (Haiti or Malawi), or continuous yearly surveys (Senegal 2012-2019). Our model provides a tool to identify the sources of variability in readiness and process quality metrics, which can help inform data collection efforts around more specific target metrics. For instance, if the target of estimation is a metric displaying strong spatial variability and little temporal variability, a less frequent but more geographically diverse sample may be appropriate. Conversely, a regularly conducted survey with smaller samples would be best suited to estimate metrics displaying substantial temporal but lower spatial variability. A previous study building upon the repeated assessment of the same facilities over two rounds of the SPA survey (2013/2014-2015/2016) in Senegal found greater variations over time in process quality compared to readiness of care metrics[31]; yet in the present study, we found substantial random temporal variations in both process and readiness of care metrics when considering all facilities assessed in each wave, which may reflect both sampling variability and changes in the quality metrics. Furthermore, as readiness and process quality metrics are commonly calculated as composite indicators of items derived from WHO guidelines, they can be flawed if any of their constituent parts are biased[39]. Change in items' definition or operationalization may lead to inconsistent estimates of single-item availability across surveys (e.g., facility access to electricity in Kenya; Supplementary Table S7). Moreover, because facility surveys assessments are designed to be representative geographically not temporally, cross-sectional analyses could miss patterns of seasonality in items' availability or protocols performed[40]. For example, in settings where malaria is endemic, depending on the timing of facility assessments, the availability of essential medicines and tests (antimalarial drugs and tests), and providers' compliance with diagnostic protocols (checking for fever) may vary substantially[41], and cloud trends in readiness and process quality metrics. Jointly analysing multiple facility surveys, through a model-based approach, can be used to investigate seasonality in items' availability and providers' compliance, and therefore test the consistency of readiness and process quality metrics.

Our study has several limitations. First, while this study utilised facility survey data collected in three African countries, its authors are predominantly based in North America, and are not experts of the local health systems or context. Thus, estimates of healthcare quality metrics presented here are meant to demonstrate potential uses of the modelling approach and should not be interpreted beyond this purpose. With the recent launch of OECD's Patient-Reported Indicator Surveys initiative[42], large, standardised, international facility assessments across European and Northern America countries will soon be available for future applications of the model. Second, the SPA and SDI exclude non-formal healthcare providers, which can comprise a substantial portion of health service delivery in many LMICs[43]. In Kenya, Senegal, and Tanzania, it was estimated however that fewer than 1% of sick children with fever, cough, or diarrhoea, were brought to traditional healers by their caretakers[44], suggesting that most sick-child consultations are treated by formal health providers. Third, we estimated process quality metrics from survey instruments using two different assessment methods (direct observation of consultations vs clinical vignettes), although previous studies have shown that there is a difference between what health providers know and what they do[45,46]. Nevertheless, in Tanzania, where data collection for SPA and SDI overlapped in 2014, estimates of process quality derived from the two assessment methods led to similar estimates (35.3% and 36.4%, respectively). Further harmonisation between data collection instruments, as entailed by WHO's Harmonised Health Facility Assessment initiative, could enhance the comparability of survey estimates. Fourth, composite healthcare quality metrics were calculated as summative measures, which assume the equal importance of different items to

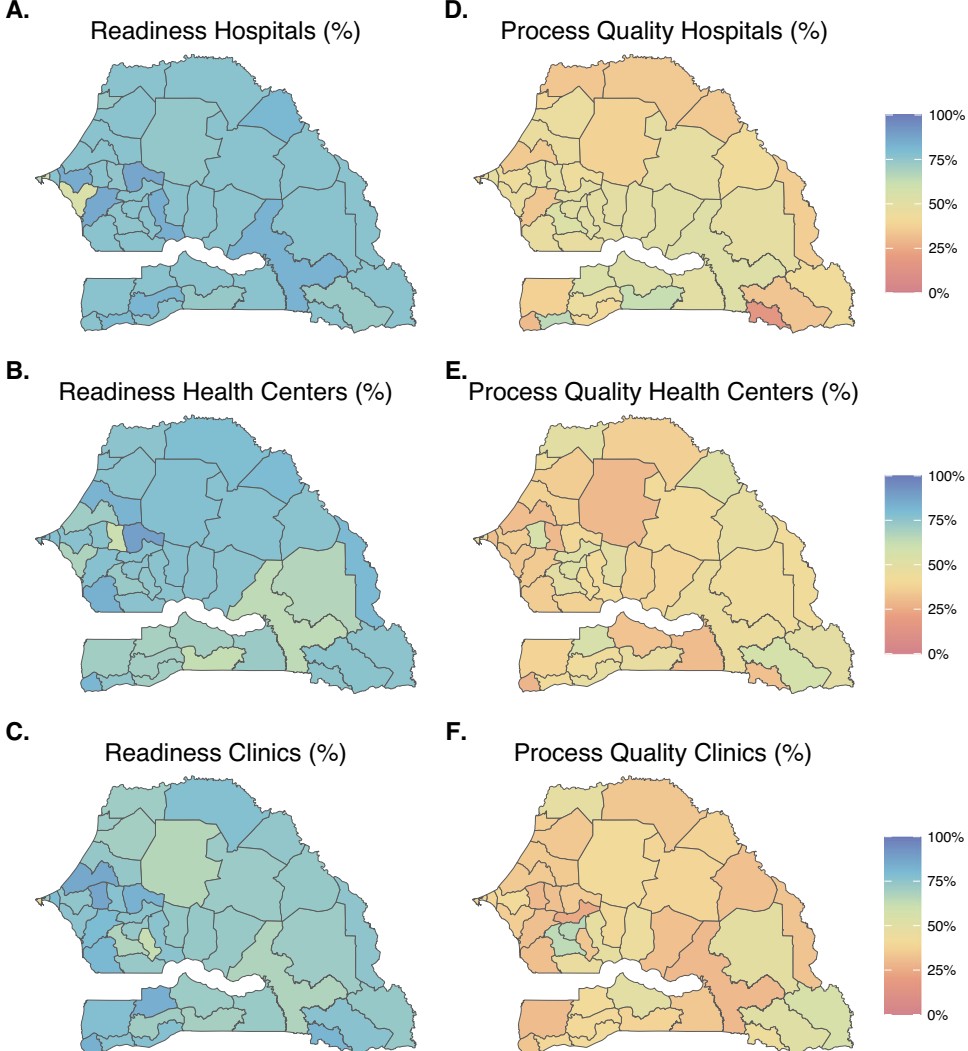

**Fig. 7 | Maps of model-estimated readiness (left panel) and process quality (right panel) metrics by subnational areas, and managing authorities, in Senegal in 2020.** Panels **A**, **B**, and **C** (respectively **D**, **E**, and **F**) are maps of modelled area-level estimates of readiness (respectively process quality) for analyses stratified on facility type. Source data are provided as a Source Data file.

quality, and ultimately to health outcomes. Items could be assigned weights based on their contribution to health improvements. However, such an approach would require equally large assumptions about the relative importance of each item, and we opted for equal weighting, the most common approach in the literature to date[47,48]. Finally, while a common technique to improve estimation, space-time smoothing can contribute to overly similar estimates among neighbouring areas and/or over time –thus masking true inequities and/or changes occurring in a given place. For instance, by smoothing trends over time, our model may mitigate a decrease observed in the survey estimates of readiness and process quality due to a 9-month national health worker strike in Senegal, which greatly overlapped with the SPA 2018 data collection estimates[49]. However, smoothing processes are useful to highlight longer-term trends to avoid the risk of over-interpretating estimates from a single survey, which are subject to temporary shocks.

Our study presents a small area model that addresses methodological gaps for analysing several facility surveys from different assessment tools. This provides a tool to contextualize the results of any single facility survey across all available surveys in a given country and enables to assess variations in healthcare quality metrics at a subnational scale and over time. This method could be applied to other contexts, areas of care, and data collection instruments.

Leveraging existing facility data can highlight gaps and challenges in health service provision and its quantification.

## Methods
### Data sources
Facility data come from two international health facility assessment tools that are publicly available and collect information on process quality: the SPA and the SDI. The SPA and SDI are standardised health facility surveys, designed to be nationally representative of the formal health sector. SPA surveys generally use a stratified survey design by facility type (e.g., hospital, health centres, clinics), managing authority (e.g., public and private), and first administrative division (or broader health zones grouping several administrative divisions), and typically include four modules: an inventory questionnaire, observations of consultations, exit interviews with the observed patients, and interviews with healthcare providers. The SDI survey is stratified by urban/rural areas and first administrative division (or broader zones), and comprises three modules: an inventory questionnaire, clinical vignettes to assess providers' knowledge, and unannounced visits to facilities to measure providers' absenteeism.

In this analysis, we used all publicly available cross-sectional facility surveys in Kenya, Senegal, and Tanzania, three countries with several nationally representative facility assessments providing

## Table 1 | List of facility surveys publicly available in Kenya, Senegal, and Tanzania and their sample size

| Country and year of surveys | Subnational unit[c] (N) | Number of facilities sampled (N) | | |
|---|---|---|---|---|
| | | All | Offering child curative services | Number of sick child consultations/vignettes |
| **Kenya** | **Counties (47)** | | | |
| SPA 1999 | | 388 | 382 | 623 |
| SPA 2004 | | 440 | 391 | 1211 |
| SPA 2010 | | 695 | 640 | 2016 |
| SDI 2012 [a] | | 294 | 294 | 625 |
| SDI 2018 | | 3094 | 3094 | 4545 |
| **Senegal** | **Departments (45[d])** | | | |
| SDI 2010[a] | | 151 | 151 | 153 |
| SPA 2012-13[b] | | 364 | 342 | 1307 |
| SPA 2014[b] | | 374 | 349 | 1213 |
| SPA 2015[b] | | 384 | 356 | 1263 |
| SPA 2016[b] | | 386 | 356 | 1029 |
| SPA 2017 | | 399 | 372 | 1064 |
| SPA 2018 | | 343 | 318 | 885 |
| SPA 2019 | | 342 | 329 | 718 |
| **Tanzania** | **Regions (26[e])** | | | |
| SPA 2006 | | 611 | 603 | 2559 |
| SDI 2010[a] | | 175 | 175 | 165 |
| SPA 2014 | | 1188 | 1154 | 4805 |
| SDI 2014 | | 403 | 403 | 570 |
| SDI 2016 | | 400 | 400 | 543 |
| **Total** | | 10,431 | 9783 | 24,976 |

[a]The 2010 SDI surveys in Senegal and Tanzania and the 2012 SDI survey in Kenya were pilot studies, which only sampled a small number of facilities in selected areas of the countries and were therefore excluded from the main analysis.
[b]Dependent sampling structure between the first four rounds of the continuous SPA SPA-survey in Senegal 2013-2016 (see Supplementary Table S1). Data from SPA 2012-13 and 2014, and from 2015 and 2016, respectively, were pooled to mitigate the effect of the dependent sampling on estimates' comparability.
[c]We indicate the subnational levels at which the analyses were conducted in this study. SPA and SDI surveys were however typically powered to produce reliable estimates at coarser levels; provinces (7) in Kenya, regions (14) in Senegal, and zones (8), regrouping several regions, in Tanzania.
[d]Although a 46th department was created in Senegal in 2021, the country was divided into forty-five departments between 2012 and 2020, when the data used in this analysis were collected.
[e]Tanzania was divided into twenty-six regions at the beginning of our study period in 2006. Although four new regions were created in 2012, and one in 2016 (such that Tanzania now counts 31 regions), we used the twenty-six-region divide in our analyses to ensure that our results would not be affected by historical boundary changes.

information on the formal health system over time, and therefore offering a unique opportunity to demonstrate the use of the small area model to estimate levels and trends in quality of care across subnational areas. We focused on child health services as they are the targets of large national and international investments in all three countries[50]. We analysed data from 10,431 facilities offering child curative care, 18,693 direct observations of sick-child consultations, and 6283 clinical vignettes, in Kenya (1999-2018), Senegal (2012-2019), and Tanzania (2006-2016) (see Table 1) to characterize facility-level information relating to the availability, readiness and process quality of child health service provision. Additionally, we used facilities' sampling weights, and the survey design variables – facilities' first administrative location, type and managing authority.

### Healthcare quality metrics
We extracted indicators reflective of general and child curative services readiness, as provided by the WHO SARA guidelines[14] and

processes of care from the Integrated Management of Childhood Illness (IMCI)[51]. The readiness metric was based on the availability and functioning of general equipment, basic amenities, staff training and supervision, essential medicines, and diagnostic capacities, specific to the provision of child curative care services. The items included in the readiness metric, listed in Supplementary Table S4, were selected based upon previous guidelines and research in paediatric quality of care[51,52]. As some of these items were not collected in the SDI surveys and older SPA surveys, we modified the metric in Kenya and Tanzania (staff training and supervision items were excluded) to ensure readiness was estimated from the same set of items across years in each country (Supplementary Table S4). The readiness metric was calculated, for each health facility offering child curative care, as the proportion of items available the day of the facility assessment.

We derived the metric of process quality of care from the content of sick child consultations observed in the SPA surveys, and the clinical knowledge displayed by health providers sampled for vignettes in the SDI surveys. Following previous studies[7,53], adherence to IMCI diagnostic protocols was used as a proxy for process quality of sick-child care. Our metric of process quality of care was calculated as the proportion of fifteen IMCI diagnostic protocols adhered to by providers during sick-child consultations or vignettes (see Supplementary Table S5).

Both metrics assume equal importance of all items and protocols to readiness and process quality, respectively, which facilitates interpretation of estimates.

### Statistical analyses
Facility survey data were complemented with covariates, which have a known or postulated relationship with health services provision. The dataset of temporally varying covariates at the subnational level included total population under five years old, travel time to the nearest settlement of more than 50,000 inhabitants, travel time to nearest health facility (walking and motorised), health worker density, urbanicity, night-time lights, average educational attainment, human development index, and elevation. We extracted secondary data to complement facility survey data, using sources like WorldPop or the Institute for Health Metrics and Evaluation Global Health Data Exchange, to identify pertinent indicators (complete list of data sources and processing in Supplementary Table S3).

The metrics of readiness and process quality of care were modelled separately using a small area estimation approach that incorporates sampling weights[54,55]. First, facility-level metrics of readiness and process quality were aggregated at the subnational area-level using facilities' sampling weights. Explicitly incorporating surveys' sampling mechanism through the use of weights is a common method for bias removal when analysing complex surveys[25–27]. Second, we estimated seven multi-level logistic models, which represented subnational area-level readiness and process quality metrics as a function of both independent and temporally structured year random effects (model 1-7), independent (model 1-4) or spatially correlated areal random effects (model 5-7), spatially and temporally indexed covariates that we hypothesised to be predictive of health service provision (model 3,4 and 7), and survey type random effects (model 2,4,6 and 7). Supplementary sections 2.2-2.5 provide a complete description of these models. While space-time random effects and covariates were used to improve the precision of our estimates by leveraging correlation structures in space, time and with auxiliary variables, survey random effects were utilised to account for systematic differences (in design or implementation) between survey instruments. For each metric, we calculated goodness of fit and model complexity indicators (LCPO, DIC, WAIC) to identify the best-performing model out of the seven models (Supplementary Section 2.5). Posterior distributions of all model parameters and hyperparameters were estimated using integrated nested Laplace approximations[56], implemented in the statistical

package R-INLA (version 22.12.16)[57] in R. We then generated and mapped annual area-level estimates of the two metrics for each Kenyan county from 1999 to 2020, each Senegalese department from 2010 to 2020, and each region of Tanzania from 2005 to 2020. Estimates and their associated 95% uncertainty intervals were obtained by drawing 1000 posterior samples for all parameters estimated in the model and calculating the mean, and the 2.5 and 97.5 quantiles.

To test the predictive validity of our models, we performed hold-one-area-out cross-validation for all areas and years where survey data were available. Hold-one-area-out involves holding out all observations in an area and year when fitting the model, and comparing the areal model's prediction with the observed direct estimate derived from the survey[32]. For each metric, we assessed the bias of our predictions by examining the mean difference between the predictions and the observed estimates (mean error), the precision of our predictions, by calculating the average distance between the predictions and the observed estimates (mean absolute error), and the calibration of our predictions by calculating the 50%, 80 and 95% coverage, i.e., the frequency at which the direct survey estimate was contained within the model's predicted 50% (respectively 80 and 95%) uncertainty interval. As facility surveys are not powered to produce reliable estimates at fine spatial resolution, benchmarking of hold-one-area-out-predictions was done against a consolidated validation set of area-year survey estimates with high-enough precision (or conversely, small-enough variance)[58], which was chosen to be estimates with precision greater than the median precision, by metric and country.

### Reporting summary
Further information on research design is available in the Nature Portfolio Reporting Summary linked to this article.

## Data availability
**Input data**: The facility-level survey data used in this analysis is publicly available from the DHS (https://dhsprogram.com/) and the World Bank's Service Delivery Indicators (https://www.sdindicators.org/) websites. Environmental covariate data were derived from high-resolution satellite imagery collected by institutions including the European Space Agency and the Defence Meteorological Satellite Program, and made publicly available by the NASA (https://modis.gsfc.nasa.gov/data/dataprod/). Socio-demographic covariate data were extracted from two research institutes: WorldPop (https://www.worldpop.org/), and the Institute for Health Metrics and Evaluation (https://ghdx.healthdata.org/). See Supplementary Table S3 for the list of covariate data and their associated source. Administrative boundaries were retrieved from the Global Administrative Unit Layers dataset and used to produce maps. **Results**: All estimates, including yearly readiness and process quality of care metrics by subnational areas with upper and lower bounds, are available in the GitHub repository described in the "Code availability" section. Data used in the figures of this publication can also be found in the GitHub repository described in the "Code availability." Source data are also provided with this paper.

## Code availability
**Data collection**: Data from DHS-SPA were downloaded manually from https://dhsprogram.com/ and cleaned using the following code: https://github.com/DHSProgram/DHS-Analysis-Code/tree/main/EffectiveCoverage. Data from World Bank's SDI were downloaded manually from https://www.sdindicators.org/ and cleaned using the following code: https://github.com/worldbank/SDI-Health. No additional software was used for data collection. **Statistical analysis**: Code used for statistical analyses and modelling is available from GitHub: https://github.com/aallorant/sae_facility_surveys[59]. This repository also contains the Source data and code used to generate the figures in this paper. All maps and figures presented in this study were generated

by the authors using R version 4.0.1. All statistical models were fitted using the R package INLA version 22.12.16.

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

## Acknowledgements

We are grateful for Dr. Emelda Okiro for her comments and suggestions, and for Dr Nathaniel Henry and Dr. Patrick Liu for conversations on health facility access modelling. Additionally, we acknowledge that the SPA and SDI surveys used in this article depended upon the dedicated efforts of the many individuals who collected the data and who worked to

assure data quality, including health workers, staff of the national statistics organisations; without their efforts, these analyses would not have been possible. A.A. is supported by a Postdoctoral Fellowship from Fonds de Recherche Santé Quebec, H.H.L reports financial support from ICF International and the World Bank, J.W. is supported by National Institutes of Health (R01AI029168), N.P. reports funding from Health Resources and Services Administration (U91HA0680), PATH, National Institutes of Health (1R21EB032229 – 01A1) and US Centers for Disease Control and Prevention (GH20-2036; CDC-RFA-GH22-2250). N.F., R.C.R,. D.P., J.L.D., M.S., D.G. and E.E. report no relevant funding related to this study.

## Author contributions

A.A., R.C.R., N.P., D.P., J.L.D., and J.W. conceived and designed the study. N.F., H.H.L., M.S., D.G. and E.E. contributed to model development and/or revisions. A.A. drafted the manuscript and all authors provided critical reviews with substantial scientific content. All authors approved the final version of the manuscript.

## Competing interests

N.F. reports financial support from Gates Ventures since June 2020 outside of the submitted work. All other authors declare no competing interests.
