## [Peer Review File · Nature Communications]

A small area model to assess temporal trends and sub-national disparities in healthcare quality using facility surveysREVIEWER COMMENTS

Reviewer #1 (Remarks to the Author):

The article is very interesting as it attempts to show that the use of SPA/SDI surveys can provide a better understanding of sub-national situations. I am not competent to judge the details of the methods used and I hope that other reviewers will be able to verify the validity of the analyses. As far as I am concerned, beyond its scientific value, this article is of methodological interest and therefore deserves to be published.

However, it seems to me that the article often goes a little beyond this desire to focus on a method to present the findings and analyse the data in the context of the three countries concerned. I, therefore, propose that the authors focus solely on methodological issues and reduce very significantly the empirical parts as they are only descriptive and take up space in the article without the authors being able to provide explanations. By refocusing on the method, the authors could better develop the potential use of their approach in the discussion.

Indeed, it is clear that the authors are not familiar with the context of the three countries (factual errors are made about decentralisation, for example), which poses a twofold problem.

Firstly, without interpretation of the findings presented, they are very limited in scope and could lead to over-interpretation. Without understanding the reasons for the results, and the policies that have been implemented over the last 10 years, the data are of little use. The authors could thus draw, for example, on the recent paper by Rudasingwa et al (DOI: 10.1186/s12939-022-01624-5) to produce a second paper that is far from methods and focused on outcomes. What can policymakers in the three countries do with the results of this paper? The question would not arise if the article were solely methodological in nature. One of the papers cited by the authors rightly complains about the limited use of research.

<https://onlinelibrary.wiley.com/doi/full/10.1111/tmi.13701>

The second problem is at the heart of current debates about epistemic injustice and the decolonisation of knowledge. How can Canadian researchers be studying data (even secondary data) from three African countries without any researchers from those countries, with the involvement of a single person from a ministry of health in those countries? In 2022, especially when one is starting a PhD. and is in training, this is no longer acceptable. The authors could, for example, refer to the reflexive approaches now being requested by journals concerned with these issues:

<https://associationofanaesthetists-publications.onlinelibrary.wiley.com/doi/full/10.1111/anae.15597>

Reviewer #2 (Remarks to the Author):

This study developed models to examine the temporal and spatial trend of quality of care (access quality and preparedness quality) using health facility surveys. It also used the models to extrapolate the quality care at the sub-national level where no quality data were available. The proposed approach was applied in Kenya, Senegal, and Tanzania. The study shows stagnation in quality of care in Kenya and Senegal, but persistent quality improvement in Tanzania. There is also significant disparity in quality of care in sub-regions in the three countries.

Understanding the status and inequity of quality of care at sub-national level is important for countries to address quality and inequity concerns. As the authors mentioned, there have been some cross-sectional studies and even longitudinal studies that were conducted using SDI and/or SPA surveys.^{1,2} The proposed model used co-variables at the sub-national level, temporal, spatial, and survey random effects, as well as interactions of space and time as well as structured spatial random effects to estimate quality of care. I appreciate the authors' effort to estimate the quality of care for places where no data were available. However, the fitted results are not stable in two countries and

sometimes do not make full sense. For example, In Kenya, the increase of readiness was minimal over the 19 years, from 1999 to 2018. There was a reduction in readiness in 2010, compared to the quality in 2004. Similarly, the readiness in Senegal was reduced between 2012 and 2019, and the process quality was instable. The instability of quality-of-care leads to doubt on

1. Construction and measurement of the quality of care. Though SDI and SPA are comprehensive surveys, it is not sure if the questions in the survey capture the essential of quality of care. Macarayan et al commented that SPA does not capture key elements of primary care quality.² Thus, using SDI and SPA to construct quality of care should be cautious.

2. Given the substantially difference of health facilities included in the study, it is not clear whether the results are comparable, particularly when examining the detailed availability tables in the appendix

- 3.1. For example, some items in 2020 in Kenya was particularly low, such as electricity, emergency transport, malaria diagnostic tools, quality insurance, ect. These items were even lower those in 1999, which draws the concerns of the comparability of data cross different years. Additionally, there were years when hospitals were not sampled, it is not clear how quality of care index was estimated at the national level.

Understanding quality of care is important, but I am not convinced that using existing facility surveys would a good approach given the validity concerns of the surveys on quality measures and results that are hard to explain from the survey.

Other methodological suggestions are:

1. It would be good provide justifications of the selection of the covariates (e.g., travel time, health worker density, education, urbanicity, ect).

2. Appendix 3.2 provide the decomposing of sources of variations; it would be good to provide overall goodness of fit indicators. Given that the models were used for extrapolation, we would expect a high level of goodness of fit.

3. In some years, no hospital and no private health facilities were sampled, please provide information on how the hospitals and private health facilities were weighted.

The discussion focuses on the explaining the advantage of the approach and explaining. However, it does not provide explanation of the trend of the quality of care, particular the drop in quality in some years. It would be more useful that the discussion could be diverted to explain the trend and validity of the approach.

References:

1. Leslie HH, Hategeka C, Ndour PI, et al. Stability of healthcare quality measures for maternal and child services: Analysis of the continuous service provision assessment of health facilities in Senegal, 2012-2018. *Trop Med Int Health* 2022; 27(1): 68-80.

2. Macarayan EK, Gage AD, Doubova SV, et al. Assessment of quality of primary care with facility surveys: a descriptive analysis in ten low-income and middle-income countries. *Lancet Glob Health* 2018; 6(11): e1176-e85.

Reviewer #3 (Remarks to the Author):

The authors present a statistical analysis of health facility-based survey data from three sub-Saharan African countries estimating two summary metrics related to sick-child healthcare over space and time. Their statistical model synthesizes two distinct but compatible surveys series in a way that accounts for each survey's unique sampling scheme. By smoothing over space, time, and survey series, the authors obtain estimates of readiness and process quality that are more precise than direct survey estimates. They identify several trends in their estimates and hypothesize about what could drive their results.

The statistical tools and methods the authors describe in the Supplemental Information are modern and robust, and appropriate for this application. Using random effects to fit to all surveys simultaneously allows each survey to contribute to the estimates according to its precision. This method reconciles the data sources' varying sample sizes and survey designs without forcing the analyst to make subjective decisions about how to weigh each source.

Based on the authors' literature review, this is the first study to model the two largest sources of data on healthcare quality simultaneously over space and time. I am not familiar with the state of the art in healthcare quality measurement, but if this is the first study to combine these data sets and model them over space and time, then it is a step forward in measuring healthcare quality. The proposed model can help measure the quality of sick-child care in any country with either of the two surveys used in this analysis. Furthermore, the ability of their model to synthesize the two survey series could offer clarity in settings where the data sources conflict with each other. The conclusions the authors draw are largely reasonable given their results, but there are number of points I would like the authors to address.

General comments

I believe the methodology is sound and consistent with modern approaches to similar applications, but I have a number of clarifying questions about methodological details and discussion points.

1. How were predictions generated in unobserved years? The authors refer several times to imputing over unobserved years and regions, but Figures 2-4 suggest that predictions were only generated in years with data. Taking Figure 4 as an example, the model seems to linearly interpolate the 95% CIs between 2006 and 2014, the first two years with data in Tanzania. Such consistent linear change over unobserved years seems impossible under an autoregressive model. This looks to me like an artefact of the default ggplot2 behavior. If those truly are the estimates for unobserved years, the authors should address the surprising consistency and precision in the discussion. On the other hand, if the authors are not estimating random effects for unobserved years, they must say so and modify Figures 2-4 so that they do not imply the existence of precise estimates where there are none.

2. Why were Kenya, Senegal, and Tanzania selected? The analysis was conducted on data from three countries in sub-Saharan Africa. Can the authors please include a comment about why they selected those three countries and how data availability in the selected countries compares to the rest of the world? Are there any biases that restricting to those particular countries could introduce?

3. Did the authors consider alternate modelling approaches? This study includes a reasonably wide model comparison experiment, but I wonder whether the authors considered alternate modelling approaches. Specifically, if they are interested in estimating metrics comprised of many components, could they have modelled each component separately in a multivariate model and then calculated the resulting metrics from posterior samples? Can they also comment on how their decision to aggregate their outcomes to the area level affects their results? Did they consider a facility-level model? The approach they used was certainly appropriate, but a brief discussion of other strategies would be informative.

4. Posterior predictive checks. The "high coverages" (line 377) the authors found in their cross-validation exercise are, in general, much higher than is desirable. Ideally, 80% posterior predictive intervals will cover 80% of the data, but the authors observed 80% posterior predictive coverage of less than 90% in only one of six country-metric combinations. The authors should not refer to "high coverage" as a strength. "Low mean squared errors" (line 377) are also difficult to interpret given that the presented MSEs are scaled. Can they please include a more interpretable error metric (unscaled RMSE on the scale of percentage points, for example) to complement their scaled MSE?

5. Additional discussion. There are few discussion points I would appreciate clarification/elaboration on:
- Apologizing in advance for my ignorance of Tanzanian history, what is the 2015 setback in Tanzania attributable to? That estimate is a large, consistent outlier. Does it match the data?
 - What do the authors mean when they say that "finding covariates...is challenging"? If this is related to the performance of covariates in their model selection process, more details about those results are necessary.
 - The explanation of increasing spatial disparity is plausible, but is it possible that this is a statistical phenomenon? If the data are lower variance in more recent years (due to larger or more efficient surveys, for example), then the model will be able to distinguish more precisely between areas. Increasing precision seems to be reflected in Kenya but not Senegal. Can the authors please briefly address circumstances that could challenge their interpretation of this key result?
6. Will code and data be made available? I was unable to determine if the authors intend to make their code and data available. At minimum, the model code should be available on GitHub, with some type of replication data (possibly synthetic).

Minor suggestions (main text)

I have a number of smaller suggestions

- Line 16: Define "readiness" and "process quality" in the Abstract
- Line 23: For the uninitiated, are all guidelines equally important? Aggregating over guidelines assumes that they are.
- Line 26: "identifies" seems extraneous in "enables identifies estimation"
- Lines 46-47: Should "including the Service Provision Assessment (SPA) and the Service Delivery Indicators (SDI)" be changed to "including the Service Provision Assessment (SPA) and the Service Delivery Indicators (SDI) surveys"?
- Line 50: I think "as" is unnecessary in "as grouped into"
- Lines 46-52: Consider including context about the global coverage of these surveys. How many LMICS conduct them? Are they only in sub-Saharan Africa?
- Lines 59-61: Please clarify this statement "Statistical modelling frameworks offer means by which such differences can be explicitly..." to give the reader some intuition about how statistical modelling can account for these differences.
- Line 72: What does "each" mean in "Our model supports the use of publicly available facility data in each country"? Each country with one or both of the surveys used here?
- Line 91: As discussed above, please specify why these three countries were selected.
- Lines 99-100: Can the authors comment on the representativeness of these surveys with respect to all facilities in these countries? This is addressed in the Discussion, but I think one sentence would be appropriate in the Methods as well.
- Line 123: Please clarify what "ensure comparability of readiness estimates" means. Does this mean calculating the proportion based on only the observed metrics?
- Line 130: Please comment on the relative importance of these 15 protocols. This is discussed briefly in the discussion, but the assumption of equal weight seems important enough to address in the Methods.
- Line 139: Please change "IHME" to "the Institute for Health Metrics and Evaluation." The acronym has not been introduced yet in the text and is only used once anyway.
- Line 151: The supplemental information indicates that the survey effects are random effects. If that is correct, please clarify in the main text because the current phrasing suggests to me that they are fixed with a base case (requiring a very different interpretation).
- "Statistical Analyses" section: Please include a brief reference to and appropriate citation for the software used to fit the models (e.g. "We fit all models with the "R-INLA" R library...").
- Line 156: Please briefly describe how goodness of fit and model complexity were measured.
- Line 164: Clarify the cross-validation strategy. Does "a given area" mean that leave-one-out cross-validation was conducted for every area in the study?
- Line 172: Please add a summary of the model selection results. As nicely addressed in the

Discussion, the relative importance of these different dimensions is interesting on its own. Which covariates were predictive?

19. Figure 1: Personally, I find the temporal trends on this plot difficult to parse. The panel for Tanzania is clear, but the cloud of points in the panel for Kenya could conceal important within-area variation. I would consider plotting change in process quality against change in readiness (measured between two years or as an average), so that each region is plotted only once.

20. Lines 225-230: This sentence ("Improvements in process...") is extremely complicated. Please consider simpler phrasing as multiple sentences.

21. Figures 2-4: I think the inset is helpful, but I find that I try to interpret the viridis colors as data, not labels. Perhaps a color scale better suited to categorical data would be more appropriate. Also, please see the prior comments about temporal interpolation.

22. Line 317: Please specify in the main text that the "intervals" are 80% credible intervals.

23. Line 348: "supports the incorporation of all available health facility survey data" is quite general. If there are other survey data sources outside of the two survey series used here, please consider a more specific phrasing.

24. Line 369: "not powered to directly provide reliable estimates." Please comment briefly on how area-level representativeness is maintained with data not designed to estimate at that level.

25. Line 370: Consider treating "data" as a plural noun: "was" to "were"

26. Line 379: Please define "effective coverage" more clearly

27. Line 410: Please replace "for only a few" with the exact number. "A few" seems unnecessarily vague.

28. Lines 425-426: I am confused about identifying "sources of variable in each readiness and process quality metric[s]". You use one metric for readiness and one metric for one metric for process quality. Is this referring to the individual metrics that make up the two aggregates or to alternate metrics for readiness and process quality? Please clarify.

Minor suggestions (supplemental information)

1. Section 2.2: Did the regression use a link function? Are negative predictions possible?

2. Table S10: Please label each component with the effect it corresponds to (i.e. time, space, space-time interaction, etc.).

3. Section 2.3: Please provide more details about the priors used here. "Spatially correlated" is vague.

Reviewer #1 (Remarks to the Author):

The article is very interesting as it attempts to show that the use of SPA/SDI surveys can provide a better understanding of sub-national situations. I am not competent to judge the details of the methods used and I hope that other reviewers will be able to verify the validity of the analyses.

As far as I am concerned, beyond its scientific value, this article is of methodological interest and therefore deserves to be published.

However, it seems to me that the article often goes a little beyond this desire to focus on a method to present the findings and analyse the data in the context of the three countries concerned. I, therefore, propose that the authors focus solely on methodological issues and reduce very significantly the empirical parts as they are only descriptive and take up space in the article without the authors being able to provide explanations. By refocusing on the method, the authors could better develop the potential use of their approach in the discussion.

Indeed, it is clear that the authors are not familiar with the context of the three countries (factual errors are made about decentralisation, for example), which poses a twofold problem.

Firstly, without interpretation of the findings presented, they are very limited in scope and could lead to over-interpretation. Without understanding the reasons for the results, and the policies that have been implemented over the last 10 years, the data are of little use. The authors could thus draw, for example, on the recent paper by Rudasingwa et al (DOI: 10.1186/s12939-022-01624-5) to produce a second paper that is far from methods and focused on outcomes. What can policymakers in the three countries do with the results of this paper? The question would not arise if the article were solely methodological in nature. One of the papers cited by the authors rightly complains about the limited use of research.

<https://onlinelibrary.wiley.com/doi/full/10.1111/tmi.13701>

We are grateful to Reviewer 1 for bringing important discussion points regarding the scope of this manuscript, local expertise, and authorship inclusion.

Our primary objective with this manuscript is to introduce a small area model, which would maximize the use of health facility assessments by enabling the estimation of healthcare quality indicators over time, and at programmatic resolution. We apply this approach to estimating indicators of quality for child health services in Kenya, Senegal, and Tanzania, to demonstrate a potential use of the model. We believe that a method paper with a “proof-of-concept” application can be more compelling/appealing to potential users. Nevertheless, we acknowledge that our language can be misleading and suggest that we are estimating indicators directly usable by the Ministries of Health of these three countries. We modified the language in several parts of the manuscript (Introduction, Results, Discussion) to clarify our intentions, and the interpretation that can be made of our results. We completely re-wrote the results section to focus on methodological considerations, and present mapped estimates as ‘Examples of model outputs’ to avoid over-interpretation of these maps. The discussion section has also

been re-focused towards the important insights that this approach applied to these health facility assessments can provide, and hints at alternative modelling approaches.

The second problem is at the heart of current debates about epistemic injustice and the decolonisation of knowledge. How can Canadian researchers be studying data (even secondary data) from three African countries without any researchers from those countries, with the involvement of a single person from a ministry of health in those countries? In 2022, especially when one is starting a PhD. and is in training, this is no longer acceptable. The authors could, for example, refer to the reflexive approaches now being requested by journals concerned with these issues: <https://associationofanaesthetists-publications.onlinelibrary.wiley.com/doi/full/10.1111/anae.15597>

Regarding authorship inclusion, we fully agree with Reviewer 1, and included a reflexivity statement in the Discussion section highlighting the limitation of having a majority of authors from the global north, writing an article using data collected in the global south (**page 21, lines 428-430**).

Our efforts to engage and include local researchers and stakeholders are described in the *Ethics inclusion form*; however, we acknowledge that true collaboration will require substantially more efforts. Our manuscript benefited from the inputs of researchers in the three countries; one researcher, acknowledged at the end of the manuscript, considered that their contribution to the analysis did not justify authorship; two researchers gave me their agreement to be listed as co-authors a few weeks after the deadline, but were added to the author list a week after submission (which is reflected in the MedRxiv submission history: <https://www.medrxiv.org/content/10.1101/2022.07.19.22276796v2>).

As noted by Reviewer 1, this study was part of a doctoral work, which has its own constraints and deadlines. Part of this doctoral work aimed at questioning the tools used to assess healthcare quality globally; this study tried to emphasize the importance of accounting for sub-national heterogeneity at the level at which care is delivered. The Service Provision Assessment and the Service Delivery Indicators surveys, which have been providing a wealth of information on health facilities, providers, and patients' satisfaction for almost 20 and 10 years, respectively, appeared as unique standardized assessment tools to investigate facility-based service provision. In contrast, OECD countries are only now launching the Patient-Reported Indicator Surveys (or PaRIS <https://www.oecd.org/health/paris/>), large, standardized facility assessments collecting information on facilities' characteristics, providers' practices, and patient-reported experiences of care. Once this dataset is available, we believe that our framework could be used to assess temporal trends and highlight subnational disparities in health service provision in OECD countries. Therefore, our work is thought of as an addition to the overall architecture of tools used to assess

healthcare quality globally (with an emphasis on variations at subnational spatial scale). We added these elements to the Discussion (**page 21, lines 432-435**).

Reviewer 1's comment on decentralization, which we assumed related to this sentence "*The devolution of health service provision from the national to the district-levels in many sub-Saharan countries has also substantially increased local governments' responsibilities in the planning and implementation of public health*", hinted at a factual error, which we interpreted as misusing the word decentralization; we modified the sentence to "devolution of health service provision" (**page 2, Line 72**).

Reviewer #2 (Remarks to the Author):

This study developed models to examine the temporal and spatial trend of quality of care (access quality and preparedness quality) using health facility surveys. It also used the models to extrapolate the quality care at the sub-national level where no quality data were available. The proposed approach was applied in Kenya, Senegal, and Tanzania. The study shows stagnation in quality of care in Kenya and Senegal, but persistent quality improvement in Tanzania. There is also significant disparity in quality of care in sub-regions in the three countries.

Understanding the status and inequity of quality of care at sub-national level is important for countries to address quality and inequity concerns. As the authors mentioned, there have been some cross-sectional studies and even longitudinal studies that were conducted using SDI and/or SPA surveys.^{1,2} The proposed model used co-variables at the sub-national level, temporal, spatial, and survey random effects, as well as interactions of space and time as well as structured spatial random effects to estimate quality of care. I appreciate the authors' effort to estimate the quality of care for places where no data were available. However, the fitted results are not stable in two countries and sometimes do not make full sense. For example, In Kenya, the increase of readiness was minimal over the 19 years, from 1999 to 2018. There was a reduction in readiness in 2010, compared to the quality in 2004. Similarly, the readiness in Senegal was reduced between 2012 and 2019, and the process quality was instable.

The instability of quality-of-care leads to doubt on

1. Construction and measurement of the quality of care. Though SDI and SPA are comprehensive surveys, it is not sure if the questions in the survey capture the essential of quality of care. Macarayan et al commented that SPA does not capture key elements of primary care quality.² Thus, using SDI and SPA to construct quality of care should be cautious.

This analysis uses similar operationalizations of indicators to standards used by WHO and other authors in the literature (see for instance, **Manuscript's References 3-8**) to approximate quality of care (which in many ways is not directly measurable). While acknowledging the limitations of the SPA, Macarayan and colleagues' study is a cross-sectional analysis of the most recent SPA surveys in 10 countries, described as "the most comprehensive standardized facility surveys available". Therefore, we believe that there is value in developing methods to optimize the use of these surveys, while acknowledging their limitations.

To address reviewer 2's concerns, we have sought to clarify what was assessed in this study- metrics of readiness and process quality- and edited language wherever possible around implying that these are directly reflective of quality of care (see for instance in Introduction, **page 2, lines 64-68**). We also discuss the limitations of using composite indicators to proxy healthcare quality metrics and highlight potential threats to internal consistency of these metrics- for instance, seasonality in the availability of certain items (**page 20, lines 414-427**).

2. Given the substantial difference of health facilities included in the study, it is not clear whether the results are comparable, particularly when examining the detailed availability tables in the appendix 3.1. For example, some items in 2020 in Kenya was particularly low, such as electricity, emergency transport, malaria diagnostic tools, quality insurance, ect. These items were even lower those in 1999, which draws the concerns of the comparability of data cross different years. Additionally, there were years when hospitals were not sampled, it is not clear how quality of care index was estimated at the national level.

We are grateful to Reviewer 2 for raising important points regarding the low availability of certain items for the Kenya 2010 survey, which allowed us to catch inconsistencies in our recoding of this survey. After carefully reviewing our codebooks and the Kenya 2010 survey report, we found that the lower availability of these items mentioned by Reviewer 2, and reported in Supplementary Table 3.1, were due to a change in the recoding of these items. Specifically, we found that: 1) availability of regular electricity or generator with fuel was 26% across facilities providing child health services (the estimate of 15% for 2010 that we initially reported in the Supplementary Table 3.1 only accounted for facilities with access to regular electricity, which was inconsistent with the definition used for previous years); 2) malaria diagnostic tools were available in 45% of facilities. We initially reported an estimate of 21% for this indicator, which corresponds to the proportion of all facilities with microscopy to diagnostic malaria (this matches the estimates reported in the survey report- malaria diagnostic tools were available in 45% of facilities, 46% of which had microscopy) and was inconsistent with the definition used for previous years (which included all type of malaria diagnostic tools); and 3) 23% of facilities providing child health services reported quality assurance activities (the estimate of

10% for 2010 reported in the Supplementary Table 3.1 only accounted for facilities that provided documentation of quality assurance activities, which was inconsistent with the definition used for previous years). We did not find any inconsistency in the recoding of the availability of emergency transport; the estimate of 10% is also consistent with the estimate reported in the SPA report (30% of facilities with an ambulance at disposal, and slightly less than half of them filled with fuel). All the survey estimates mentioned above can be found in the SPA 2010 official report, here: <https://dhsprogram.com/pubs/pdf/SPA17/SPA17.pdf>

We fully agree with Reviewer 2 regarding some variability in items' availability from survey to survey. We believe that this is a feature of survey data, including the most widely used household survey data, such as the Demographic and Health Survey. For instance, Burstein et al. (Manuscript's reference 18), Maheu-Giroux et al., (https://journals.lww.com/aidsonline/Fulltext/2019/12153/National_HIV_testing_and_diagnosis_coverage_in.7.aspx) and Mercer et al. (Manuscript's reference 54) all reported on conflicting trends in survey estimates of child mortality, HIV testing, and family planning indicators, respectively, and argued that model are useful, precisely to reconcile conflicting trends seen in survey estimates, and help frame our understanding of overall trends in demographic and health indicators.

We are grateful to Reviewer 2 for raising his concerns regarding the sampling of hospitals and private facilities- this was due to a formatting error for table S1, when uploading the word document. We updated the table in **Supplementary Materials section 1.1**: hospitals and private health facilities were sampled in every survey included in this analysis (we excluded the two pilot SDI surveys from the analysis precisely because their sample of health facilities was not representative of the health system).

Understanding quality of care is important, but I am not convinced that using existing facility surveys would a good approach given the validity concerns of the surveys on quality measures and results that are hard to explain from the survey.

We share Reviewer 2's concerns regarding the validity of composite metrics that are used to proxy something as complex as health service provision- especially given the variability of some of the items used to calculate these metrics. Nevertheless, these metrics are commonly used by authors in the healthcare quality measurement literature, and by health officials for funding decisions as part of results-based financing studies (see Manuscript's Reference 53). By looking jointly at multiple surveys over time, our approach can be used to critically assess these commonly used metrics, for instance, by pointing out their inconsistencies over time.

Other methodological suggestions are:

1. It would be good provide justifications of the selection of the covariates (e.g., travel time, health worker density, education, urbanicity, ect).

We included available covariates that we hypothesized to be predictive of health service provision- proximity to urban centres, indicators of poverty, development and education, concentration of health workers, etc. Availability of georeferenced auxiliary data over time was a substantial constraint, and we had to choose among those commonly used in the literature (**see for instance, Manuscript's References 16-18**), which we thought would be most relevant to health service provision. Nevertheless, the small area model does not require that all the included covariates would be good predictors of healthcare quality metrics, but only that among them, some would be informative of the spatial and/or temporal patterns in these indicators. We discuss these points **page 19, lines 384-388**.

Ultimately, including covariates only improved the models (according to the goodness of fit and model complexity criteria DIC-WAIC-LCPO) in half of the cases (process quality metric in Kenya and both metrics in Tanzania).

2. Appendix 3.2 provide the decomposing of sources of variations; it would be good to provide overall goodness of fit indicators. Given that the models were used for extrapolation, we would expect a high level of goodness of fit.

We are grateful for reviewer 2's excellent suggestion to include visualizations of the models' fit. We present in-sample validation results by country and metric with **Figure 2 on page 7**; this figure allows to discuss important features of the model, including the varying precision/reliability of direct survey estimates at subnational levels, and how the model handles estimates with large variance.

3. In some years, no hospital and no private health facilities were sampled, please provide information on how the hospitals and private health facilities were weighted.

Regarding the weighting of facilities in the analysis, we used a two-stage modelling approach that incorporates sampling weights for all facilities at the first stage of the model (presented in **Supplementary Material section 2.2**). Given the complex sampling designs of the SPA and SDI surveys, incorporating sampling weights, which are derived from the sampling frame, is a way to explicitly accounts for the substantial variations in the probability of inclusion of health facilities of different types (hospitals vs clinics, public vs private, etc.).

The discussion focuses on the explaining the advantage of the approach and explaining. However, it does not provide explanation of the trend of the quality of care, particular the drop in quality in some years. It would be more useful that the discussion could be diverted to explain the trend and validity of the approach.

We expanded the Discussion section to include a paragraph tackling the question of the validity of composite indicators used to proxy healthcare quality metrics- we highlighted potential threats to internal consistency of these metrics- for instance, seasonality in the availability of certain items (**page 20, lines 414-427**). We also discussed extensively alternative to the modelling approach presented here (**page 18, lines 357-374**).

Reviewer #3 (Remarks to the Author):

Summary

The authors present a statistical analysis of health facility-based survey data from three sub-Saharan African countries estimating two summary metrics related to sick-child healthcare over space and time. Their statistical model synthesizes two distinct but compatible surveys series in a way that accounts for each survey's unique sampling scheme. By smoothing over space, time, and survey series, the authors obtain estimates of readiness and process quality that are more precise than direct survey estimates. They identify several trends in their estimates and hypothesize about what could drive their results. The statistical tools and methods the authors describe in the Supplemental Information are modern and robust, and appropriate for this application. Using random effects to fit to all surveys simultaneously allows each survey to contribute to the estimates according to its precision. This method reconciles the data sources' varying sample sizes and survey designs without forcing the analyst to make subjective decisions about how to weigh each source. Based on the authors' literature review, this is the first study to model the two largest sources of data on healthcare quality simultaneously over space and time. I am not familiar with the state of the art in healthcare quality measurement, but if this is the first study to combine these data sets and model them over space and time, then it is a step forward in measuring healthcare quality. The proposed model can help measure the quality of sickchild care in any country with either of the two surveys used in this analysis. Furthermore, the ability of their model to synthesize the two survey series could offer clarity in settings where the data sources conflict with each other. The conclusions the authors draw are largely reasonable given their results, but there are number of points I would like the authors to address.

Comments

General comments

I believe the methodology is sound and consistent with modern approaches to similar applications, but I have a number of clarifying questions about methodological details and discussion points.

1. How were predictions generated in unobserved years? The authors refer several times to imputing over unobserved years and regions, but Figures 2-4 suggest that predictions were only generated in years with data. Taking Figure 4 as an example, the model seems to linearly interpolate the 95% CIs between 2006 and 2014, the first two years with data in Tanzania. Such consistent linear

change over unobserved years seems impossible under an autoregressive model. This looks to me like an artefact of the default ggplot2 behavior. If those truly are the estimates for unobserved years, the authors should address the surprising consistency and precision in the discussion. On the other hand, if the authors are not estimating random effects for unobserved years, they must say so and modify Figures 2-4 so that they do not imply the existence of precise estimates where there are none.

Predictions were generated for all areas, in observed and unobserved years, by drawing 1,000 posterior samples for all parameters estimated in the model presented in Supplementary Material (line 152). Figures 2-4 juxtapose two types of estimates at two different spatial resolutions: 1) direct survey estimates at the spatial resolution for which these surveys were powered for (provinces for Kenya in Fig. 2), and 2) modelled estimates at the finer spatial resolution (counties for Kenya in Fig. 2). Exactly as reviewer 3 noted, the upper panel of these figures present linear interpolation of direct survey estimates (which are only available when surveys were conducted) from ggplot2. However, we realize that our attempt to compare two types of estimates at two types of spatial resolutions on a single figure was confusing (and even misleading for the time trend of the upper panel) for the reader. We deleted these figures from the revised manuscript to reflect the reviewer's 3 excellent comment.

2. Why were Kenya, Senegal, and Tanzania selected? The analysis was conducted on data from three countries in sub-Saharan Africa. Can the authors please include a comment about why they selected those three countries and how data availability in the selected countries compares to the rest of the world? Are there any biases that restricting to those particular countries could introduce?

We thank reviewer 3 for this comment and added in the Introduction a sentence to present the geographic scope of the SPA and SDI surveys (**page 2 lines 58-59**), and the justification for these three countries (**page 3, lines 115-116**).

Kenya, Senegal, and Tanzania have conducted several rounds of health facility assessments, representative of their national health system, from both the SPA and SDI assessment tools, and as such represented unique settings to demonstrate our modelling approach. Specifically, we thought that it would be particularly informative to use as case studies countries that have adopted different strategies for health system measurement; in Kenya, facility surveys have been conducted every five years or so since 1999, while Senegal has engaged in a continuous yearly survey since 2012. More generally, there is a critical need to assess the optimal frequency and scope of health facility assessments, which so far have been conducted as occasional surveys (Kenya and Tanzania), one-time census (Haiti or Malawi), or continuous yearly survey (Senegal since 2012). Our modelling approach can be used as a first step to adequately appraise variability in healthcare quality metrics and attune data

efforts: with strong spatial variability and little temporal variability of a quality metric a less frequent but more geographically diverse sample could increase precision, while frequent but smaller samples would be more appropriate for metrics displaying substantial temporal but low spatial variability.

SPA and SDI surveys have been conducted in 17 and 14 countries respectively, so far.

Several other countries, such as the Democratic Republic of Congo, Uganda, Malawi, have conducted multiple facility surveys using the SPA or SDI collection instruments, and/or WHO's Service Availability and Readiness Assessment. The World Bank has additionally conducted large health facility assessments as part of its results-based financing programs in countries such as the Central African Republic and Cameroon. With the World Health Organization promoting a new Harmonized Health Facility Assessment (<https://www.who.int/data/data-collection-tools/harmonized-health-facility-assessment/introduction>), to ensure the alignment of health facility survey instruments and enable comparability of results over time and across countries, and OECD countries launching the Patient-Reported Indicator Surveys (or PaRIS <https://www.oecd.org/health/paris/>), we expect that our small area approach will be applicable in an increasing number of settings.

3. Did the authors consider alternate modelling approaches? This study includes a reasonably wide model comparison experiment, but I wonder whether the authors considered alternate modelling approaches. Specifically, if they are interested in estimating metrics comprised of many components, could they have modelled each component separately in a multivariate model and then calculated the resulting metrics from posterior samples? Can they also comment on how their decision to aggregate their outcomes to the area level affects their results? Did they consider a facility-level model? The approach they used was certainly appropriate, but a brief discussion of other strategies would be informative.

We considered the option of modelling each item/protocol separately and combining posterior samples to calculate final metrics, mentioned by reviewer 3. We decided to model the metrics directly (i.e. the aggregation of the items) mostly for two reasons: 1) as our conversations with stakeholders had made it clear that health facility assessments were not used for logistical purposes (stock-out of one item), we thought it would be clearer to model the metrics at the level at which they would use (i.e. as aggregated indicators); 2) modelling items separately and combining posterior samples would have increased the computational cost quite significantly, as the number of models fitted would have effectively been multiplied by the number of items/protocols.

Additionally, the facility-level model was the first modelling option we considered. Specifically, we envisioned a facility-level that would account for the survey design explicitly by including stratifying variables (typically, administrative subdivision, facility's managing authority, and facility type, e.g., hospitals/clinics/health centres) as

covariates (rather than using survey weights). However, since we aimed at providing a broader analytic approach for generating area-level estimates, this facility-level model added the difficulty of aggregation. To aggregate from facility-level estimates to area-level estimates, we would have needed to know the underlying distribution of health facilities in each area by managing authority and type (to account for the fact some types of facilities are much more common than others)- which amounts to having at disposal the original sampling frames that contain the proportions of public/private and hospital/health centres/clinics, at the spatial resolution of interest, which are typically unavailable, although some information was available in the case of the SPA surveys (in the reports). Ideally, a geo-located census of all facilities over time could be used for aggregation. Although several groups are currently working on building such a resource, it does not currently exist; existing databases of health facilities tend to miss large numbers of facilities and/or over-represent public/larger health facilities (see for instance **Manuscript's References 33 and 34**).

We opted for the area-level model because it is a modelling approach that directly incorporates survey weights (at the first stage of the model), which ensures that the survey design is acknowledged.

We are grateful to Reviewer 3 for the excellent suggestion of explicating alternative modelling options - we added a paragraph to the Discussion section (**page 18, lines 355-374**) that covers the points made above.

4. Posterior predictive checks. The “high coverages” (line 377) the authors found in their cross-validation exercise are, in general, much higher than is desirable. Ideally, 80% posterior predictive intervals will cover 80% of the data, but the authors observed 80% posterior predictive coverage of less than 90% in only one of six country-metric combinations. The authors should not refer to “high coverage” as a strength. “Low mean squared errors” (line 377) are also difficult to interpret given that the presented MSEs are scaled. Can they please include a more interpretable error metric (unscaled RMSE on the scale of percentage points, for example) to complement their scaled MSE?

As reviewer 3 stresses out, inaccurate coverage can happen with these models, and is undesirable. We modified the text to avoid ambiguous language. We also included 50% and 95% coverages to assess calibration at different nominal levels. We modified the measures of bias and precision to be more directly interpretable- we used the mean error and mean absolute error. All these changes are reflected **on page 9, with Figure 3**.

5. Additional discussion. There are few discussion points I would appreciate clarification/elaboration on:

a. Apologizing in advance for my ignorance of Tanzanian history, what is the 2015 setback in Tanzania attributable to? That estimate is a large, consistent outlier. Does it match the data?

As reviewer 3 mentioned above, the upper panel of figures 2-4 displays the direct survey estimates, with linear interpolation between survey years. The drop in both readiness and process quality observed on the top panel of figure 4 reflects the difference in direct survey estimates of readiness and process quality between the SDI surveys of 2014 and 2016, and the SPA survey of 2014-15. This drop may reflect some of the differences between the two data collection instruments- differences that are accounted for in the model, with random effects by survey type.

b. What do the authors mean when they say that “finding covariates...is challenging”? If this is related to the performance of covariates in their model selection process, more details about those results are necessary.

We included available covariates that we hypothesized to be predictive of health service provision- proximity to urban centres, indicators of poverty, development and education, concentration of health workers, etc. Availability of georeferenced auxiliary data over time was a substantial constraint, and we had to choose among those commonly used in the literature (see for instance, Manuscript’s References 16-18), which we thought would be most relevant to health service provision. Nevertheless, the small area model does not require that all the included covariates would be good predictors of healthcare quality metrics, but only that among them, some would be informative of the spatial and/or temporal patterns in these indicators. We discuss these points **page 19, lines 384-388**.

c. The explanation of increasing spatial disparity is plausible, but is it possible that this is a statistical phenomenon? If the data are lower variance in more recent years (due to larger or more efficient surveys, for example), then the model will be able to distinguish more precisely between areas. Increasing precision seems to be reflected in Kenya but not Senegal. Can the authors please briefly address circumstances that could challenge their interpretation of this key result?

We fully agree with reviewer 3’s comment- sample sizes were consistent over time in Senegal, but in Kenya, the most recent survey (SDI 2018) sampled 4 to 10 times as many facilities compared to previous surveys. As a result, enhanced precision could indeed give the impression of larger inequalities. Following the editor and reviewers’ advice, we refocused the manuscript around methodological aspects of the manuscript, and therefore deleted the substantive interpretation of estimates, including the comment regarding increasing spatial disparity.

6. Will code and data be made available? I was unable to determine if the authors intend to make their code and data available. At minimum, the model code

should be available on GitHub, with some type of replication data (possibly synthetic).

We added a **Code Availability** section (Lines 554-557); the code is now available here: https://github.com/aallorant/sae_facility_surveys.

We are extremely grateful to reviewer 3's thorough review of the manuscript- their minor suggestions were integrated directly into the revised manuscript. We responded to some of Reviewer's 3 open-ended questions in the text below.

Minor suggestions (main text)

I have a number of smaller suggestions

1. Line 16: Define "readiness" and "process quality" in the Abstract

Given the format of Nature paper, we applied this suggestion by defining readiness and process quality in the Introduction, **page 2, line 64-68.**

2. Line 23: For the uninitiated, are all guidelines equally important? Aggregating over guidelines assumes that they are.

It is indeed the implicit assumption made when using a summative measure for readiness and process quality- we discuss this point in the **Discussion page 20, lines 414-427** and **page 22, lines 448-453.**

3. Line 26: "identifies" seems extraneous in "enables identifies estimation"

This was corrected.

4. Lines 46-47: Should "including the Service Provision Assessment (SPA) and the Service Delivery Indicators (SDI)" be changed to "including the Service Provision Assessment (SPA) and the Service Delivery Indicators (SDI) surveys"?

This was modified (**page 2, line 55**)

5. Line 50: I think "as" is unnecessary in "as grouped into"

This was corrected.

6. Lines 46-52: Consider including context about the global coverage of these surveys. How many LMICS conduct them? Are they only in sub-Saharan Africa?

We now mention the coverage of these surveys in the **Introduction page 2, line 58-59.**

7. Lines 59-61: Please clarify this statement “Statistical modelling frameworks offer means by which such differences can be explicitly...” to give the reader some intuition about how statistical modelling can account for these differences.

We now expand on this statement in the **Introduction section, page 3, line 109-113.**

8. Line 72: What does “each” mean in “Our model supports the use of publicly available facility data in each country”? Each country with one or both of the surveys used here?

We referred to each of the three countries included in the analyses.

9. Line 91: As discussed above, please specify why these three countries were selected.

As mentioned above, this is explicated in the Introduction; we also explicated this choice in Methods, **page 23, lines 481-485.**

10. Lines 99-100: Can the authors comment on the representativeness of these surveys with respect to all facilities in these countries? This is addressed in the Discussion, but I think one sentence would be appropriate in the Methods as well.

Representativeness of these surveys is described in Methods section, **page 23, line 471-472**

11. Line 123: Please clarify what “ensure comparability of readiness estimates” means. Does this mean calculating the proportion based on only the observed metrics?

We modified the text to make it explicit that we we modified the readiness metric in Kenya and Tanzania (staff training and supervision items were excluded) to only include items collected in both SPA and SDI surveys, **page 25, lines 537-540.**

12. Line 130: Please comment on the relative importance of these 15 protocols. This is discussed briefly in the discussion, but the assumption of equal weight seems important enough to address in the Methods.

We added a sentence **page 26, lines 568-569.**

13. Line 139: Please change “IHME” to “the Institute for Health Metrics and Evaluation.” The acronym has not been introduced yet in the text and is only used once anyway.

We changed the text.

14. Line 151: The supplemental information indicates that the survey effects are random effects. If that is correct, please clarify in the main text because the current phrasing suggests to me that they are fixed with a base case (requiring a very different interpretation).

We modified the text to clearly reflect the fact that we were using (sum-to-one) random effects, **page 27, lines 595-596.**

15. “Statistical Analyses” section: Please include a brief reference to and appropriate citation for the software used to fit the models (e.g. “We fit all models with the “R-INLA” R library...”).

We added a reference to the section, **page 27, lines 603-605.**

16. Line 156: Please briefly describe how goodness of fit and model complexity were measured.

We present the measures of goodness of fit **lines page 27, 599-601, and in Supplemental Material, page 8, section 2.5.**

17. Line 164: Clarify the cross-validation strategy. Does “a given area” mean that leave-one-out cross-validation was conducted for every area in the study?

We now expanded on the cross-validation strategy in the Methods section, **page 28, lines 616-630.**

18. Line 172: Please add a summary of the model selection results. As nicely addressed in the Discussion, the relative importance of these different dimensions is interesting on its own. Which covariates were predictive?

We present the model selection results in **Supplementary Material, Section 3.3, page 12, and the effect of covariates on Figure S4, page 18.**

19. Figure 1: Personally, I find the temporal trends on this plot difficult to parse. The panel for Tanzania is clear, but the cloud of points in the panel for Kenya could conceal important within-area variation. I would consider plotting change in process quality against change in readiness (measured between two years or as an average), so that each region is plotted only once.

As mentioned above, we completely rewrote the Results section- this figure is not included in the revised version.

20. Lines 225-230: This sentence (“Improvements in process...”) is extremely complicated. Please consider simpler phrasing as multiple sentences.

This sentence is not included in the revised version.

21. Figures 2-4: I think the inset is helpful, but I find that I try to interpret the viridis colors as data, not labels. Perhaps a color scale better suited to categorical data would be more appropriate. Also, please see the prior comments about temporal interpolation.

These figures are not included in the revised version.

22. Line 317: Please specify in the main text that the “intervals” are 80% credible intervals.

We specified the nominal levels of each credible interval used.

23. Line 348: “supports the incorporation of all available health facility survey data” is quite general. If there are other survey data sources outside of the two survey series used here, please consider a more specific phrasing.

We changed the sentence, **page 17, line 330.**

24. Line 369: “not powered to directly provide reliable estimates.” Please comment briefly on how area-level representativeness is maintained with data not designed to estimate at that level.

The raw direct survey estimates are not powered to provide reliable estimates at the district-level; by exploiting space-time smoothing, albeit at the cost of introducing some bias, we reduce variance (and therefore uncertainty intervals) greatly. We would however argue that the bias does not make the estimates less representative as smoothing models are advantageous in terms of having lower mean squared error than direct survey estimates.

25. Line 370: Consider treating “data” as a plural noun: “was” to “were”

We modified the text.

26. Line 379: Please define “effective coverage” more clearly

The revised version does not mention effective coverage.

27. Line 410: Please replace “for only a few” with the exact number. “A few” seems unnecessarily vague.

We modified the text.

28. Lines 425-426: I am confused about identifying “sources of variable in each readiness and process quality metric[s]”. You use one metric for readiness and one metric for one metric for process quality. Is this referring to the individual metrics that make up the two aggregates or to alternate metrics for readiness and process quality? Please clarify.

This was an ambiguous wording indeed- the readiness and process quality metrics always refer to the aggregated sums of items and are modelled separately.

Minor suggestions (supplemental information)

1. Section 2.2: Did the regression use a link function? Are negative predictions possible?

The outcomes were modelled on the logit-scale using the gaussian family and back-transformed on the probability-scale using the inverse-logistic function ($f(x) = \exp(x)/(1+\exp(x))$). Therefore, negative predictions were not possible.

2. Table S10: Please label each component with the effect it corresponds to (i.e. time, space, space-time interaction, etc.).

We modified the table (now **S12**) in supplementary material page 13, line 542-547, to include a description of each component.

3. Section 2.3: Please provide more details about the priors used here. “Spatially correlated” is vague.

We provided a description of the spatial smoothing process used in the model with **supplementary Section 2.3, lines 243-251**.

REVIEWER COMMENTS

Reviewer #2 (Remarks to the Author):

1. I am not sure if the authors have fully addressed my previous concerns about the construction of QoC index. The added sentences in lines 64–68 are more about the concept of QoC, which I do not doubt. The example that I gave before on the reduction of readiness in Senegal between 2012 and 2019 leads me to be concerned about the validity of the QoC index used in the paper. First, I would appreciate the authors' assessment of whether the proposed methods could mitigate the impact of such type of validity issue; second, if this type of problem cannot be addressed using the proposed method, how such type of concern would affect your results?

2. The authors acknowledged inconsistent definitions of some items in the survey, such as access to electricity ect. Even with 26% access to electricity in Kenya in 2010, it is still much lower than that in 1999. It is counterintuitive. I do not think this could be explained as a feature of survey data. I cannot find any languages on how this concern was addressed in the updated manuscript. Country-year fixed effects models may help address this issue if the problem occurs in specific countries for particular years. By the way, the article did not justify why random effects models were used.

Reviewer #3 (Remarks to the Author):

The authors present a small-area statistical model to estimate readiness and process quality from two survey series that are available in a number of LMICs. The manuscript is a major revision of their previous submission, focusing more closely on the utility of their method and less on a close interpretation of their current results. I think that the reframed paper is an improvement over the original, and I commend the authors for the amount of work they have put in during the review process.

My original feedback was primarily about the Results and Discussion sections, which have been replaced almost entirely. The specific points that were still relevant from my review have been satisfactorily addressed. Their statistical method has not changed, so my assessment is the same. If they are the first authors to use a hierarchical Bayesian model to synthesize these two datasets, then the method presented here is a step forward. The tools they use are modern and appropriate for the application.

REVIEWER COMMENTS

Reviewer #2 (Remarks to the Author):

1. I am not sure if the authors have fully addressed my previous concerns about the construction of QoC index. The added sentences in lines 64—68 are more about the concept of QoC, which I do not doubt. The example that I gave before on the reduction of readiness in Senegal between 2012 and 2019 leads me to be concerned about the validity of the QoC index used in the paper. First, I would appreciate the authors' assessment of whether the proposed methods could mitigate the impact of such type of validity issue; second, if this type of problem cannot be addressed using the proposed method, how such type of concern would affect your results?

We are grateful to reviewer 2 for further explicating their previous concerns regarding the readiness and process of care quality metrics used to approximate quality of care. First, regarding the reduction in the readiness metric in Senegal that reviewer 2 is referring to, we want to highlight that this decrease is minimal and not statistically significant: estimates of readiness nationally were 65.1% (64.3-65.9%) for 2012-14, 65.7% (64.5-66.9%) for 2015-16, 66.9% (65.5-68.3%) for 2017, 65.0% (59.6-70.4%) for 2018, and 64.6% (61.3-67.9%) for 2019. The impression of a decrease may have come from the misleading upper panel of a figure included in the first version of the manuscript, showing direct survey regional-level estimates of readiness, and interpolation between these estimates. That figure was removed from the next versions of the manuscript following reviewer 3 recommendation.

Nevertheless, it remains that this figure suggested a decrease in readiness in 2018, in the western part of the country, including Dakar. As mentioned in the Discussion section, a 9-month national health worker strike in Senegal overlapped and disrupted the SPA 2018 data collection; the strike motto was precisely 'data retention' (<https://dhsprogram.com/pubs/pdf/SPA32/SPA32.pdf> p.11). Differential uptake of the mobilization by facility type and region may have introduced differential bias in the estimates by region, as public hospitals in urban centres such as Dakar were more likely to be affected by this social movement.

Second, while such disruptions in data collection can introduce bias in estimates, we believe that our proposed methods can mitigate the importance of these biases. The rationale for the framework we presented in this paper is precisely to provide a tool for funders and policymakers to contextualize the results of a survey across all available facility surveys; thus mitigating the risk of over-interpreting estimates from a single survey, which can often suggest large increase or decrease in a given metric of interest.

We propose to model time with both random walk process and zero-mean random effects. The former penalizes large differences in the metrics from one year to the following, reflecting our assumption that readiness and process quality metrics change smoothly over time. The latter is used to absorb temporary 'shocks', which explain deviations from the smooth trend introduced by the random walk.

A drawback of this choice for modelling time that we highlighted in the limitations paragraph of the discussion section is that real temporary disruptions to health services tend to be underestimated, as the model assumes continuity over abrupt changes. Yet, as explained in the previous paragraph, we justify the use of smoothing processes to model the effect of time as our goal is to provide a picture of the longer-term trends rather than the year-to-year variations.

2. The authors acknowledged inconsistent definitions of some items in the survey, such as access to electricity ect. Even with 26% access to electricity in Kenya in 2010, it is still much lower than that in 1999. It is counterintuitive. I do not think this could be explained as a feature of survey data. I cannot find any languages on how this concern was addressed in the updated manuscript. Country-year fixed effects models may help address this issue if the problem occurs in specific countries for particular years. By the way, the article did not justify why random effects models were used.

We are grateful to reviewer 2 for highlighting the inconsistency in the access to electricity item used in the calculation of the readiness metric. We agree with reviewer 2 that looking at the survey mean item availability (as presented in table S7) suggests that availability of regular electricity or generator with fuel in 2010 (26%) is inconsistent with the levels suggested with the 1999 SPA survey (58%), the 2004 SPA survey (47%), and the 56% in the SDI 2020. One possible explanation for this discrepancy is the fact that the definition of this item changed from “electricity *routinely* available during service hours or a backup generator with fuel” to a more stringent definition of “*regular, uninterrupted* electricity or a functioning generator with fuel” (<https://dhsprogram.com/pubs/pdf/SPA17/SPA17.pdf> page 35). We believe that the 2010 definition is more stringent and may have therefore been met by fewer health facilities. We added a sentence in the manuscript to underscore occurrences of these inconsistencies in single-item availability (p20, lines 315-318).

This example of an inconsistency in a specific item reinforces the views of several stakeholders we talked to (country health officials, DHS, colleagues) that these health facility assessments are not meant to be used for logistical purposes (i.e., stock-out of an item). Health information systems, such as DHIS2, are more adequate to assess the facility-level availability of specific items over time. Health facility assessments can be used to construct aggregated metrics, such as the WHO-supported metrics of readiness and process quality used in this manuscript, which are proxy or tools to highlight trends and contextualize new data collection needs and new survey estimates.

Regarding reviewer 2 suggestion of adding country-year fixed effects to account for inconsistencies between surveys, we agree that this could be a great approach; it is however incompatible with our small area approach. Following recommendations from several articles in the literature, we decided to fit separate models in each country (Li et al., Plos One, 2019, discusses the use of separate country models to fit subnational

models). Therefore, year fixed effects in a single country model would lead to identification issues. Instead, time was modelled using zero-mean independent random effects and random walk processes, as described above. We added a sentence in the Discussion section justifying the use of a smoothing process over time (p.22, lines 357-362).

Reviewer #3 (Remarks to the Author):

The authors present a small-area statistical model to estimate readiness and process quality from two survey series that are available in a number of LMICs. The manuscript is a major revision of their previous submission, focusing more closely on the utility of their method and less on a close interpretation of their current results. I think that the reframed paper is an improvement over the original, and I commend the authors for the amount of work they have put in during the review process.

My original feedback was primarily about the Results and Discussion sections, which have been replaced almost entirely. The specific points that were still relevant from my review have been satisfactorily addressed. Their statistical method has not changed, so my assessment is the same. If they are the first authors to use a hierarchical Bayesian model to synthesize these two datasets, then the method presented here is a step forward. The tools they use are modern and appropriate for the application.

We are grateful to reviewer 3 for their comments and feedback.

REVIEWERS' COMMENTS

Reviewer #2 (Remarks to the Author):

The revision is fine with me now.